


**Coccolithophore populations and their contribution to carbonate export during an**
**annual cycle in the Australian sector of the Antarctic Zone**
Andrés S. Rigual Hernández[1,*], José A. Flores[1], Francisco J. Sierro[1], Miguel A. Fuertes[1],
Lluïsa Cros [2] and Thomas W. Trull[3,4]
1 Área de Paleontología, Departamento de Geología, Universidad de Salamanca, 37008
Salamanca, Spain
2 Institut de Ciències del Mar, CSIC, Passeig Marítim 37-49, 08003 Barcelona, Spain.
3 Antarctic Climate and Ecosystems Cooperative Research Centre, University of
Tasmania, Hobart, Tasmania 7001, Australia
4 CSIRO Oceans and Atmosphere Flagship, Hobart, Tasmania 7001, Australia
*corresponding author
Email: arigual@usal.es
**Abstract**
The Southern Ocean is experiencing rapid and relentless change in its physical and
biogeochemical properties. The rate of warming of the Antarctic Circumpolar Current
exceeds that of the global ocean, and the enhanced uptake of carbon dioxide is causing
basin-wide ocean acidification. Observational data suggest that these changes are
influencing the distribution and composition of pelagic plankton communities. Long-term
and annual field observations on key environmental variables and organisms are a critical
basis for predicting changes in Southern Ocean ecosystems. These observations are
particularly needed, since high-latitude systems have been projected to experience the
most severe impacts of ocean acidification and invasions of allochthonous species.
Coccolithophores are the most prolific calcium carbonate producing phytoplankton
group, playing an important role in Southern Ocean biogeochemical cycles. Satellite
imagery has revealed elevated particulate inorganic carbon concentrations near the major
circumpolar fronts of the Southern Ocean, that can be attributed to the coccolithophore
*Emiliania huxleyi*. Recent studies have suggested changes during the last decades in the
distribution and abundance of Southern Ocean coccolithophores. However, due to limited
field observations, the distribution, diversity and state of coccolithophore populations in
the Southern Ocean remain poorly characterized.



We report here on seasonal variations in the abundance and composition of
coccolithophore assemblages collected by two moored sediment traps deployed in the
deep ocean (~2000 and 3700 m) in the Australian sector of the Antarctic Zone for one
year in 2001-02. Additionally, seasonal changes in coccolith weights of *E. huxleyi*
populations were estimated using circularly polarized micrographs analysed with *C-*
*Calcita* software. Our findings indicate that (1) coccolithophore sinking assemblages
were nearly monospecific for *Emiliania huxleyi* morphotype B/C in the Antarctic Zone
waters in 2001-2002; (2) coccolith assemblages experienced weight and length reduction
during the summer months; (3) the estimated annual coccolith weight of *E. huxleyi* at both
sediment trap depths (2.11 ± 0.96 and 2.13 ± 0.90 pg at 2000 m and 3700 m) was
consistent with previous studies for morphotype B/C in other Southern Ocean settings;
(4) coccolithophores accounted for approximately 2-5% of the annual, deep-ocean $CaCO_3$
flux. Our results are the first annual record of coccolithophore abundance, composition
and degree of calcification in the Antarctic Zone. They provide a baseline against which
to monitor coccolithophorid responses to changes in environmental conditions expected
for this region in coming decades.

**Key words**: Southern Ocean, Antarctic Zone, coccolithophores, coccolith weight,
sediment traps.

**Introduction**
The rapid increase in atmospheric $CO_2$ levels since the onset of the industrial
revolution is modifying the environmental conditions of marine ecosystems in a variety
of ways. The enhanced greenhouse effect, mainly driven by increased atmospheric $CO_2$
levels, is causing ocean warming (Barnett et al., 2005), shallowing of mixed layer depths
(Levitus et al., 2000) and changes in light penetration and nutrient supply (Bopp et al.,
2001; Rost and Riebesell, 2004; Sarmiento et al., 2004b; Deppeler and Davidson, 2017).
Moreover, the enhanced accumulation of $CO_2$ in the ocean is giving rise to changes in the
ocean carbonate system, including reduction of carbonate ion concentrations and
lowering of seawater pH. Most evidence suggests that the ability of many marine
calcifying organisms to form carbonate skeletons and shells may be reduced with
increasing seawater acidification including some species (but not all) of coccolithophores,
corals, pteropods and foraminifera (e.g. Orr et al., 2005; Moy et al., 2009; Lombard et al.,





2010; Beaufort et al., 2011; Andersson and Gledhill, 2013). Since phytoplankton are
extremely sensitive to global environmental change (Litchman et al., 2012) all predicted
changes in marine environmental conditions are likely to modify the abundance,
composition and distribution of phytoplankton communities.
Changes in the relative abundances of major phytoplankton functional groups are
likely to influence ocean biogeochemistry and ocean carbon storage, with feedbacks to
the rate of climate change (e.g. Boyd and Newton, 1995; Boyd et al., 1999; Falkowski et
al., 2004; Cermeño et al., 2008). For example, diatoms can play a prominent role in export
of organic matter from the surface ocean, because of their heavy siliceous frustules and
capacity for aggregation and rapid sinking facilitates efficient transport of organic carbon
(Buesseler, 1998; Smetacek, 1999), although this silica-mediated carbon export may not
reach the ocean interior efficiently (Francois et al., 2002; Lam and Bishop, 2007). The
precipitation and sinking of $CaCO_3$ by coccolithophores also has the potential for
complex contributions to carbon cycling.  Carbonate precipitation removes more
alkalinity than dissolved inorganic carbon from surface waters, thereby acting to increase
$pCO_2$ in surface waters (the so-called carbonate counter pump, e.g.  (Zeebe, 2012), but
ballasting by carbonates appears to increase transfer of organic carbon to the ocean
interior (Armstrong et al., 2002; Klaas and Archer, 2002). On seasonal timescales the
counter pump contribution dominates (Boyd and Trull, 2007), but more complex
interactions can occur over longer timescales as a result of changing extents of carbonate
dissolution in sediments, including the possibility that enhanced calcite dissolution in the
Southern Ocean contributed to lower atmospheric $CO_2$ levels during glacial maxima
(Archer and Maier-Reimer, 1994; Sigman and Boyle, 2000; Ridgwell and Zeebe, 2005).
The Southern Ocean is a critical component of the Earth's ocean–climate system
and plays a pivotal role in the global biogeochemical cycles of carbon and nutrients
(Sarmiento et al., 2004a; Anderson et al., 2009). Despite its relatively small area (~25%
of the global ocean), the Southern Ocean contains ~40% of the global ocean inventory of
anthropogenic $CO_2$ (Khatiwala et al., 2009; Takahashi et al., 2009; Frölicher et al., 2015),
and it exports nutrients to more northern latitudes ultimately supporting ~ 75% of the
ocean primary production north of 30°S (Sarmiento et al., 2004a). Model projections
suggest that the reduction in the saturation state of $CaCO_3$ will reach critical thresholds
sooner in cold, high-latitude ecosystems such as the Southern Ocean (Orr et al., 2005;
McNeil and Matear, 2008; Feely et al., 2009). Therefore, calcifying organisms living in
these regions will be the first to face the most severe impacts of ocean acidification.





In view of the rapid changes in climate and other environmental stressors presently
occurring in the Southern Ocean, a major challenge facing the scientific community is to
predict how phytoplankton communities will reorganize in response to global change. In
this regard, two main aspects of the distributions of coccolithophores are emerging.
Firstly, coccolithophores are dominantly present in the Subantarctic Southern Ocean, a
feature termed by Balch et al. (2011) as the "Great Calcite Belt" based on satellite
reflectance estimates of PIC abundances. Although importantly the PIC accumulations
are significantly less than those that arise in the North Atlantic, and the satellite algorithm
is not reliable in Antarctic waters, where it badly overestimates PIC abundances (Balch
et al., 2016; Trull et al., 2017). Secondly, recent studies suggest that the magnitude and
geographical distribution of *E. huxleyi* blooms may be experiencing significant and rapid
changes. Cubillos et al. (2008) and Winter et al. (2014) postulated that *E. huxleyi* has
expanded its ecological niche south of the Polar Front in the recent decades.
Contrastingly, Freeman and Lovenduski (2015) suggested an overall decline in Southern
Ocean PIC concentrations using satellite records between 1998 and 2014. The
explanation of these contrasting results may lie in the methodologies applied. While
shipboard surface water observations provide a highly detailed picture of a given
ecosystem, they are very spares, only represent a snapshot in time, and can easily miss
blooms of any given species. The satellite PIC signal has the great advantage of largescale
and repeated coverage, but can miss subsurface populations (e.g. Winter et al., 2014) and
be mimicked by the spectral characteristics of other scattering sources, such as
microbubbles (Zhang et al., 2002), glacial flour (Balch et al., 2011) and noncalcifying
organisms such as *Phaeocystis antarctica* (Winter et al., 2014), a colonial
prymnesiophyte algae very abundant in high latitude systems of the Southern Ocean (e.g.
Arrigo et al., 1999; Arrigo et al., 2000). Notably the PIC algorithm performs particularly
poorly in Antarctic waters (Balch et al., 2016; Trull et al., 2017)
For these reasons, year-round field observations of areas representative of key
Southern Ocean regions are essential to determine the current state of coccolithophore
communities and to develop baselines against which long-term trends can be detected.
Moreover, a better understanding of coccolithophore distribution, ecology and seasonal
dynamics is required to improve our interpretations of the sedimentary record and our
models of biogeochemistry. Sediment traps are a direct method to collect data about
calcareous and siliceous micro and nanoplankton. Traps allow the monitoring of seasonal
and annual variability of plankton export, document species successions, and help to



determine the specific role of microplankton species in the biological and carbonate
pumps. The autonomous collection capacity of sediment traps is particularly useful in the
remote Southern Ocean, where inaccessibility and harsh working conditions prevent year-
round ship-based sampling.

We present here the first record of composition, abundance, and seasonality of

coccolithophore assemblages in the Antarctic Zone of the Southern Ocean, a record
inferred from one-year records from two deep ocean sediment traps deployed on a single
mooring south of Australia at the site of the SOIREE ocean iron fertilisation experiment
near 61°S, 140°E (Boyd et al., 2000a). Moreover, we report weight and length
measurements on *E. huxleyi* coccoliths, assessing the impact of seasonally varying
environmental parameters on *E. huxleyi* coccoliths. That provides a baseline of coccolith
dimensions for the populations living in this region. All the above information is needed
for monitoring coccolithophore responses, if any, to changing environmental conditions
in the Antarctic Zone south of Australia during coming decades.

**2. Material and Methods**
**2.1 Regional setting and oceanography**

The southern Antarctic Zone (AZ-S; Parslow et al., 2001) is delimited at the north

by the southern branch of the Polar Front (PF) and at the south by the southern front of
the Antarctic Circumpolar Current (SAACF). Trull et al. (2001b) summarized the
seasonal evolution of water column properties in the study region. The intense heat loss
of surface waters during winter decreases Sea Surface Temperature (SST) to values <
1°C, resulting in strong vertical convection. Winter mixing extends to depths of about
120 m, replenishing the upper water column with nutrients. Chlorophyll-*a* levels during
winter are negligible throughout the region due to the reduced solar radiation and the
deep, continuous vertical mixing. During summer, increasing solar radiation warms the
surface ocean and a seasonal thermocline forms (Fig. 2). By late summer (March) SST
ranges between 2 and 3 °C. Considerable nutrient depletion associated with a moderate
increase in algal biomass occurs within the mixed layer. Nonetheless, due to the limited
sampling of the study region, the timing of the summer nutrient minimum is not well
constrained by the available data (Trull et al., 2001b). Silicate exhibits the strongest
summer draw-down of all the macronutrients, reaching ~30% of its winter values (Fig. 2;
Trull et al., 2001), mainly due to diatom growth and subsequent biogenic silica export to



the deep sea (Rigual-Hernández et al., 2015a). The low algal biomass accumulation in the
region is attributed to the very low iron levels (0.1-0.2 nM; Boyd et al., 2000a; Sohrin et
al., 2000). Mesozooplankton analysis during the SOIREE experiment by Zeldis (2001)
indicates that zooplankton community in the study region is dominated by copepods,
mainly large calanoid copepodites. Grazing pressure was low (<1% of the phytoplankton
standing stock removed per day) and, therefore, is thought not to play an important role
in the control of the micro-phytoplankton (primarily diatom) stocks, but nanoflagellate
grazer abundances were significant and were likely to have regulated smaller
phytoplankton abundances (Hall and Safi, 2001).

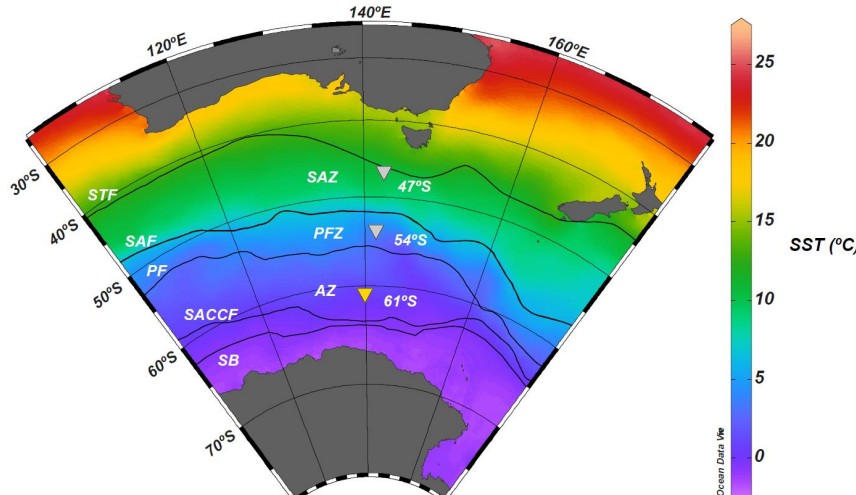


**Figure 1**. Annual mean sea surface temperature map (World Ocean Atlas; Locarnini et
al., 2013) of the Australian sector of the Southern Ocean, showing the position of the
main frontal and zonal systems (adapted from Orsi et al., 1995) and the location of the
61°S, 54°S and 47°S sediment trap stations (inverted triangles). Abbreviations: STF -
Subtropical Front, SAZ - Subantarctic Zone, SAF - Subantarctic Front, PFZ - Polar
Frontal Zone, PF - Polar Front, AZ - Antarctic Zone, SACCF - Southern ACC Front and
SB - Southern Boundary.






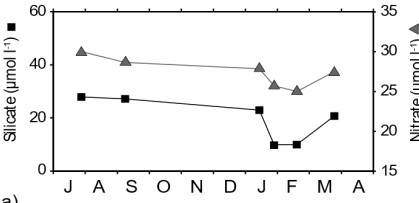 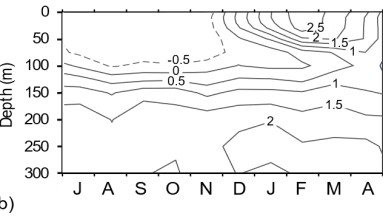

a)                        b)

**Figure 2**: (a) Summary of seasonal evolution of macronutrient concentrations (silicate
and nitrate) at the 61°S site taken from the WOCE SR3 transects between 1993 and 1996
(modified from Trull et al., 2001b) (b) Seasonal variation in the vertical structure of
temperature (°C) between June 2001 and April 2002 for the 61°S site from the World
Ocean Atlas 2009 (Locarnini et al., 2010).

**2.2 Water carbonate chemistry**

Calcite solubility increases at higher pressures and lower temperatures, so that

dissolution increases with depth in the water column. Based on downward changes in the
calcite dissolution rate, two critical depth horizons can be distinguished: the calcite
saturation horizon (CSH) that can be defined as the depth at which the water becomes
undersaturated with respect to calcite (i.e. where $\Omega_{calcite} = 1$); and the $CaCO_3$
compensation depth (CCD), the depth at which the rate of calcite rain from the upper
water column equals the dissolution rate. Figure 3 shows carbonate concentrations $[CO_3^{2-}$
] and calcite saturation ($\Omega_{calcite}$) for the WOCE SR03 2001 transect between Antarctica
and Tasmania along the 140°E meridian as estimated by Bostock et al. (2011). In the AZ-
S waters south of Tasmania, the CSH and CCD occur at 3000 and 3700 m, respectively
(Fig. 3). Therefore, the location of sediment traps at the 61°S site allows for the
assessment of dissolution changes, if any, of coccolithophore assemblages between the
two critical dissolution depth horizons: the CSH and CCD. Notably, both progressive
uptake of anthropogenic $CO_2$ and increased upwelling of naturally $CO_2$ rich deep waters
over the past 20 years is leading to shallowing of these features (Pardo et al., 2017)






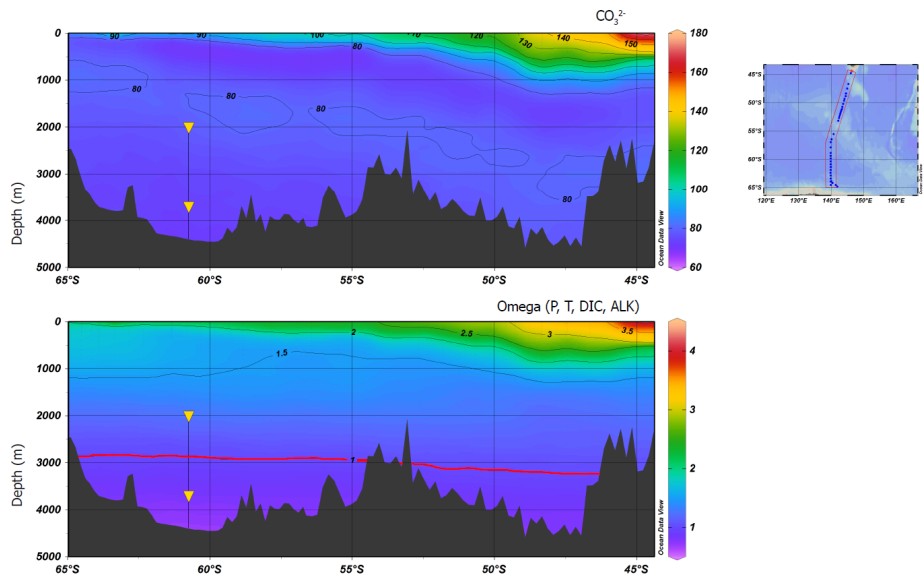

**Figure 3.** Cross section of the mooring location (only 2000m and 3700m sediment traps
are represented) in comparison to regional seafloor bathymetry, carbonate concentrations
[$CO_3^{2-}$] and calcite saturation ($\Omega_{calcite}$) for WOCE transect SR03 2001 from Bostock et al.
(2011), who calculated them from the DIC and alkalinity in the CARINA database
(Tanhua et al., 2008 ). The location of the transects is shown on the map on the right top.
$\Omega_{calcite}$ = 1 contour is highlighted with a red line to show the approximate depth of the
CSH across the transect.

### 2.3 Sediment trap experiment

As part of the SAZ collaborative research program (Trull et al., 2001c), a sediment

trap experiment was carried out at the 61°S site (60° 44.43´S; 139° 53.97´E) in the
Australian sector of the southern Antarctic Zone within the region where the Southern
Ocean Iron Release Experiment (SOIREE) was conducted (Boyd et al., 2000a). The 61°S
site is characterized by weak currents with a mean eastward geostrophic surface velocity
of approximately $0.03 \pm 0.02$ m s$^{-1}$ (Trull et al., 2001b). The site is north of the Seasonal
Sea-Ice Zone (Massom et al., 2013; Rigual-Hernández et al., 2015a) and remote from any
known iceberg pathway (Gladstone et al., 2001).

The 61°S mooring was equipped with three McLane Parflux time series sediment

traps (Honjo and Doherty, 1988) for approximately one year (November 30, 2001 to
September 29, 2002, 317 days). The traps were located at approximately 1000, 2000 and





3700 m below the surface in a water column of 4393 m; Figure 3b). Each trap was
provided with 21 cups. Sampling intervals were synchronized between traps and in order
to resolve the seasonal flux cycle ranged from 8 days (in austral summer) to 55 days in
austral winter. No samples were recovered from the shallowest trap owing to equipment
malfunction and, therefore, only results for the 2000 and 3800 m traps are presented here.
Each trap was paired with an Aanderaa current meter and temperature sensors. The 250
ml collection cups were filled with a buffered solution of sodium tetraborate (1 g $L^{-1}$),
sodium chloride (5 g $L^{-1}$), strontium chloride (0.22 g $L^{-1}$), and mercury chloride (3 g $L^{-1}$)
in unfiltered, deep seawater from the region. The two deeper traps completed their
collection sequence as programmed, providing continuous time-series for a year. Due to
the low particle fluxes during the winter, insufficient material remained for phytoplankton
analysis of cup 1 from the 2000 m trap and cups 1, 2, 19, 20 and 21 from the 3700 m trap
(Table 1).

**2.4 Sample processing and coccolithophore counting procedure**

The sediment trap cup contents were washed through a 1 mm sieve after recovery

and then divided into 10 aliquots using a rotary splitter (McLane, Inc.). A description of
the analytical procedures for estimation of geochemical fluxes is provided in Trull et al.
(2001a) and Rigual-Hernández et al. (2015a). One aliquot was used for siliceous and
calcareous micro- and nano-plankton analyses.  Each fraction for plankton analysis was
refilled with distilled water to 40 ml, from which 10 ml was subsampled and buffered
with a solution of sodium carbonate and sodium hydrogen carbonate (pH 8) and kept
refrigerated for calcareous nannoplankton analysis. Samples for coccolithophore analysis
were prepared following the methodology of Flores and Sierro (1997). In short, 300 μl
were extracted with a micropipette and dropped onto a glass Petri dish previously filled
with a buffered solution and with a cover slip on its bottom. After settling 12 hours, the
buffer solution was removed using short strips of filter paper placed at the edge of the
dish. Then, the cover slip was left to dry completely and mounted on a glass slide using
Canada balsam. Coccoliths were identified and counted using a Nikon Eclipse 80i
polarized light microscope at 1000× magnification. A minimum of 400 coccoliths were
counted in each sample. Coccospheres occurred in much lower numbers than loose
coccoliths in these preparations. The coccolith counts were transformed into daily fluxes
of specimens $m^{-2}$ $d^{-1}$ following the formula:




$$F = \frac{N \times \dfrac{A}{n \times a} \times V \times S}{d \times T}$$



where "*F*" is the daily coccolith flux, "*N*" the number of coccoliths, "*A*" the total area of
a Petri dish, "*n*" the number of fields-of-view analysed, "*a*" the area of a single field of
view, "*V*" the dilution volume, "*S*" the split of the cup, "*d*" the number of days of
collection and "*T*" the aperture area of the sediment trap.

Since the sediment trap collection period was shorter than a full calendar year, an

estimate of the annual coccolith flux of the 2000 m trap was calculated. This estimate
takes into account the fact that the unsampled days occurred in winter when particle fluxes
were low, and were obtained by using the flux for the last winter cup (#21 in 2002) to
represent mean daily fluxes during the unobserved interval. Due to the lack of samples
corresponding to the winter 2002 for the 3700 m sediment trap record, the annualization
of the coccolith fluxes for this trap was made based only on the samples with available
data. Therefore, the annualized and annual flux data for the 3700 m trap presented in
Table 1 should be used with caution.



| 61_2000 Cup | Sampling period mid point | Length days | Total Mass Flux mg m$^{-2}$ d$^{-1}$ | CaCO$_3$ mg m$^{-2}$ d$^{-1}$ | CaCO$_3$ % | POC mg m$^{-2}$ d$^{-1}$ | POC % | Diatoms 10$^6$ valves m$^{-2}$ d$^{-1}$ | Coccolithophore flux 10$^8$ coccoliths m$^{-2}$ d$^{-1}$ | E. huxleyi | C. leptoporus | Other |
|---|---|---|---|---|---|---|---|---|---|---|---|---|
| 1 | nov. 30, 2001 | 8 | 48 | 14 | 30 | 0.7 | 1.5 | - | - | - | - | - |
| 2 | dic. 08, 2001 | 8 | 78 | 17 | 22 | 1.7 | 2.2 | 9 | 2.5 | 98.8 | 1.2 | 0.0 |
| 3 | dic. 16, 2001 | 8 | 326 | 62 | 19 | 6.9 | 2.1 | 82 | 2.7 | 98.5 | 1.3 | 0.2 |
| 4 | dic. 24, 2001 | 8 | 509 | 140 | 28 | 6.4 | 1.3 | 85 | 8.2 | 99.5 | 0.5 | 0.0 |
| 5 | ene. 01, 2002 | 8 | 1151 | 44 | 4 | 26.9 | 2.3 | 408 | 12.3 | 99.8 | 0.2 | 0.0 |
| 6 | ene. 09, 2002 | 8 | 1069 | 170 | 16 | 14.8 | 1.4 | 200 | 22.3 | 99.8 | 0.2 | 0.0 |
| 7 | ene. 17, 2002 | 8 | 656 | 60 | 9 | 11.3 | 1.7 | 159 | 9.2 | 99.3 | 0.7 | 0.0 |
| 8 | ene. 25, 2002 | 8 | 702 | 38 | 5 | 11.0 | 1.6 | 296 | 8.4 | 99.3 | 0.7 | 0.0 |
| 9 | feb. 02, 2002 | 8 | 666 | 39 | 6 | 12.0 | 1.8 | 184 | 5.4 | 98.8 | 1.2 | 0.0 |
| 10 | feb. 10, 2002 | 8 | 595 | 24 | 4 | 8.2 | 1.4 | 295 | 6.0 | 99.5 | 0.5 | 0.0 |
| 11 | feb. 18, 2002 | 8 | 534 | 20 | 4 | 6.2 | 1.2 | 149 | 9.8 | 99.0 | 0.5 | 0.5 |
| 12 | feb. 26, 2002 | 8 | 524 | 19 | 4 | 4.7 | 0.9 | 152 | 5.0 | 100.0 | 0.0 | 0.0 |
| 13 | mar. 06, 2002 | 8 | 586 | 15 | 3 | 6.9 | 1.2 | 120 | 6.4 | 99.8 | 0.2 | 0.0 |
| 14 | mar. 14, 2002 | 8 | 285 | 11 | 4 | 3.2 | 1.1 | 71 | 2.0 | 99.8 | 0.2 | 0.0 |
| 15 | mar. 22, 2002 | 8 | 290 | 7 | 3 | 3.2 | 1.1 | 66 | 2.0 | 97.6 | 1.0 | 1.5 |
| 16 | mar. 30, 2002 | 8 | 263 | 8 | 3 | 2.6 | 1.0 | 87 | 0.9 | 99.2 | 0.8 | 0.0 |
| 17 | abr. 08, 2002 | 10 | 264 | 7 | 3 | 2.2 | 0.8 | 97 | 1.3 | 98.1 | 1.9 | 0.0 |
| 18 | may. 08, 2002 | 50 | 130 | 5 | 4 | 1.2 | 1.0 | 47 | 0.8 | 99.8 | 0.2 | 0.0 |
| 19 | jun. 29, 2002 | 54 | 65 | 2 | 4 | 0.7 | 1.0 | 10 | 0.7 | 98.8 | 0.8 | 0.4 |
| 20 | ago. 22, 2002 | 55 | 56 | 2 | 4 | 0.8 | 1.5 | 19 | 0.9 | 99.5 | 0.2 | 0.2 |
| 21 | sep. 29, 2002 | 20 | 42 | 2 | 4 | 0.5 | 1.3 | 6 | 0.9 | 98.0 | 2.0 | 0.0 |
| Annualised values | | | 232 | 17 | 7.4 | 3.3 | 1.4 | 67 | 2.8 | | | |
| Annual flux | | | 85 g m$^{-2}$ y$^{-1}$ | 6 g m$^{-2}$ y$^{-1}$ | | 1.2 g m$^{-2}$ y$^{-1}$ | | 24 10$^9$ valves m$^{-2}$ y$^{-1}$ | 1.03 10$^{11}$ coccoliths m$^{-2}$ y$^{-1}$ | 99.4 | 0.5 | 0.1 |

| 61_3700 Cup | Sampling period mid point | Length days | Total Mass Flux mg m$^{-2}$ d$^{-1}$ | CaCO$_3$ mg m$^{-2}$ d$^{-1}$ | CaCO$_3$ % | POC mg m$^{-2}$ d$^{-1}$ | POC % | Diatoms 10$^6$ valves m$^{-2}$ d$^{-1}$ | Coccolithophore flux 10$^7$ coccoliths m$^{-2}$ d$^{-1}$ | E. huxleyi | C. leptoporus | Other |
|---|---|---|---|---|---|---|---|---|---|---|---|---|
| 1 | nov. 30, 2001 | 8 | 38 | 9 | 23 | 0.4 | 1.1 | - | - | - | - | - |
| 2 | dic. 08, 2001 | 8 | 31 | 9 | 28 | 0.4 | 1.2 | - | - | - | - | - |
| 3 | dic. 16, 2001 | 8 | 99 | 29 | 30 | 1.4 | 1.4 | 4 | 1.3 | 99.0 | 0.7 | 0.2 |
| 4 | dic. 24, 2001 | 8 | 231 | 59 | 26 | 1.4 | 0.6 | 12 | 5.5 | 99.3 | 0.5 | 0.2 |
| 5 | ene. 01, 2002 | 8 | 873 | 87 | 10 | 17.3 | 2.0 | 118 | 11.6 | 99.8 | 0.2 | 0.0 |
| 6 | ene. 09, 2002 | 8 | 1157 | 154 | 13 | 19.8 | 1.7 | 479 | 15.9 | 100.0 | 0.0 | 0.0 |
| 7 | ene. 17, 2002 | 8 | 828 | 166 | 20 | 9.4 | 1.1 | 354 | 20.0 | 100.0 | 0.0 | 0.0 |
| 8 | ene. 25, 2002 | 8 | 490 | 34 | 7 | 6.4 | 1.3 | 169 | 11.0 | 99.8 | 0.2 | 0.0 |
| 9 | feb. 02, 2002 | 8 | 491 | 32 | 6 | 6.5 | 1.3 | 385 | 4.6 | 100.0 | 0.0 | 0.0 |
| 10 | feb. 10, 2002 | 8 | 419 | 19 | 4 | 6.0 | 1.4 | 281 | 4.2 | 99.8 | 0.2 | 0.0 |
| 11 | feb. 18, 2002 | 8 | 584 | 36 | 6 | 6.2 | 1.1 | 254 | 15.9 | 99.1 | 0.7 | 0.2 |
| 12 | feb. 26, 2002 | 8 | 581 | 31 | 5 | 5.2 | 0.9 | 238 | 12.2 | 100.0 | 0.0 | 0.0 |
| 13 | mar. 06, 2002 | 8 | 849 | 23 | 3 | 7.6 | 0.9 | 326 | 15.0 | 99.8 | 0.2 | 0.0 |
| 14 | mar. 14, 2002 | 8 | 369 | 18 | 5 | 3.3 | 0.9 | 44 | 6.6 | 99.2 | 0.8 | 0.0 |
| 15 | mar. 22, 2002 | 8 | 218 | 8 | 4 | 2.6 | 1.2 | 32 | 6.6 | 99.5 | 0.2 | 0.2 |
| 16 | mar. 30, 2002 | 8 | 258 | 10 | 4 | 2.5 | 1.0 | 43 | 6.8 | 99.3 | 0.7 | 0.0 |
| 17 | abr. 08, 2002 | 10 | 257 | 9 | 3 | 2.3 | 0.9 | 32 | 4.8 | 99.5 | 0.2 | 0.2 |
| 18 | may. 08, 2002 | 50 | 118 | 5 | 4 | 1.2 | 1.0 | 8 | 1.2 | 99.8 | 0.0 | 0.2 |
| 19 | jun. 29, 2002 | 54 | 0 | 0 | 4 | 0.0 | 1.0 | - | - | - | - | - |
| 20 | ago. 22, 2002 | 55 | 0 | 0 | 4 | 0.0 | 1.0 | - | - | - | - | - |
| 21 | sep. 29, 2002 | 20 | 0 | 0 | 4 | 0.0 | 1.0 | - | - | - | - | - |
| Annualised values | | | 188 | 17 | 9 | 2.3 | 1.2 | 62 | 3.3 | | | |
| Annual flux | | | 69 g m$^{-2}$ y$^{-1}$ | 6 g m$^{-2}$ y$^{-1}$ | | 0.9 g m$^{-2}$ y$^{-1}$ | | 23 10$^9$ valves m$^{-2}$ y$^{-1}$ | 1.20 10$^{11}$ coccoliths m$^{-2}$ y$^{-1}$ | 99.7 | 0.2 | 0.1 |

**Table 1**: Daily export fluxes of total mass flux, calcium carbonate (CaCO$_3$), particulate organic carbon (POC), diatom valves and coccoliths registered at the 61°S site from November 2001 through October 2002. Mass fluxes listed as zero were too small to measure (<1 mg).

### 2.5 SEM analysis

As the resolution of the light microscope is insufficient to differentiate *Emiliania huxleyi* morphotypes, the samples of the 2000 m trap record were analysed using Scanning Electron Microscopy. Glass cover-slips were prepared following the decantation method outlined by Flores and Sierro (1997). The dried cover-slips were mounted on aluminium stubs and coated in gold. A EVO HD25 SEM (Carl Zeiss) was used to determine the morphotype of *Emiliania huxleyi* coccoliths found in the samples. Due to the large abundance of diatom valves and the scarcity of coccoliths in the samples, a compromise between number of identified coccoliths and time spent had to be reached. Therefore, a target minimum of thirty *Emiliania huxleyi* coccoliths per sample were identified. The taxonomic concepts of Young and Westbroek (1991), Young et al. (2003),





Cubillos et al. (2007) and Hagino et al. (2011) were followed to classify the *Emiliania*
*huxleyi* coccoliths into morphotypes.

## 2.6 C-Calcita analyses

The glass slides used for coccolith counts were also analysed for coccolith mass
and size measurements using a with a Nikon Eclipse LV100 POL polarized light
microscope equipped with circular polarization and a Nikon DS-Fi1 8-bit colour digital
camera. Calibration images were performed on an apical rhabdolith of the genus
*Acanthoica* collected by a sediment trap at the 47°S site (46°48′S, 142°6′E), located in
the Australian sector of the Subantarctic Zone. Camera parameters and microscope light
settings were maintained constant throughout the imaging session. Depending on
coccolith concentration, between 13-28 random fields of view per sample were
photographed. The images were then analysed by the image processing software *C-*
*Calcita* (Fuertes et al., 2014). The output files for single *E. huxleyi* coccoliths were
visually selected. Length and weight measurements were automatically performed by *C-*
*Calcita* software. A total of 2328 coccoliths were analysed with a minimum of 50
coccoliths per sample. For more methodological details see Fuertes et al. (2014).
An estimated range of annual contributions of coccoliths to total $CaCO_3$ export
was calculated for the 2000 m trap record by multiplying the coccolith flux of each
sampling interval by the maximum and minimum standard deviations of coccolith weight
values measured on each sample. Then, the minimum and maximum estimates of
coccolith-$CaCO_3$ fluxes for each sampling interval (i.e. cup) were used to estimate the
minimum and maximum annual contribution of coccoliths to total carbonate following
the same procedure as for the annual coccolith fluxes.

## 2.7 Satellite imagery, meteorological and oceanographic data

Weekly mean sea surface temperatures (SST) for the 2001-2002 interval were
obtained from the NOAA Optimum Interpolation Sea Surface Temperature Analysis
database (Reynolds et al., 2002). Seasonal SST variation range was low, with maximum
SSTs of 2.94 °C observed during March 2002 and minimum of 0.12 °C, in early October
2002. SST variations mirrored changes in the vertical structure of the water column
temperature profile (Fig. 4) that displayed vertical homogeneity of the water column in
autumn and winter and a seasonal thermocline during the austral summer (Fig. 2b).




Photosynthetically active radiation (PAR), monthly chlorophyll-*a* concentration
and particulate inorganic carbon (PIC) concentration estimates were obtained from
NASA's Giovanni program (Acker and Leptoukh, 2007) (Fig. 4) for the region: 130°E,
62.5°S, 150°E, 59.5°S. Chlorophyll-*a* concentration was low throughout the year (ranging
from 0.07 to 0.30 mg m$^{-3}$) and in line with previous observations in the study region (Trull
et al., 2001b). Algal biomass responded rapidly to the solar radiation increase in
September 2001 and reached its highest levels in November 2001 (Fig. 4). Chlorophyll-
*a* concentration declined throughout the summer, reaching negligible values in autumn
and winter (i.e. from March to August 2002). Satellite-derived PIC concentration
exhibited a clear seasonal pattern similar to that of the chlorophyll-*a* with peak
concentrations in November (up to 0.003 mol m$^{-3}$) and values below detection limit in
winter (Fig. 4).



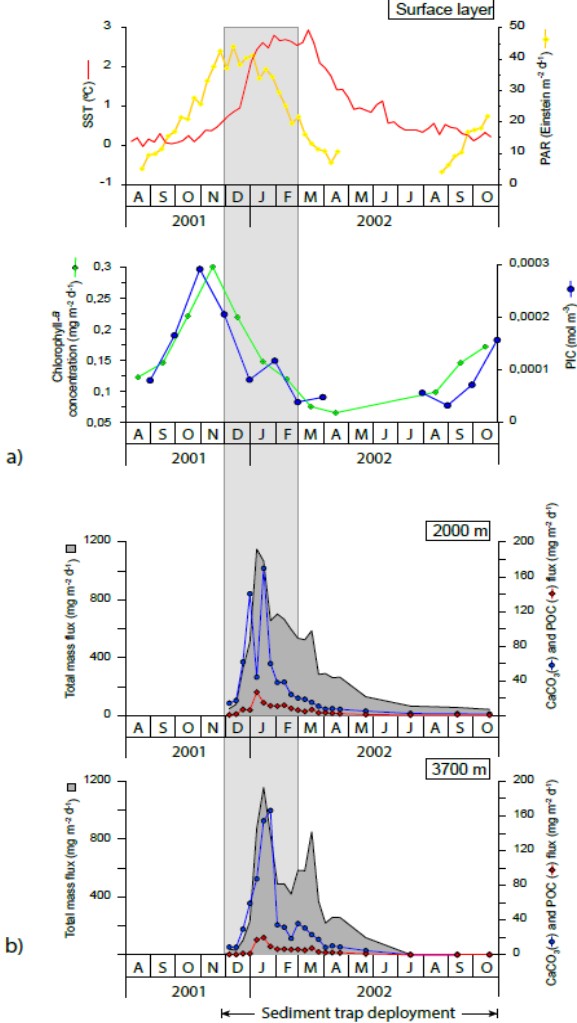


**Figure 4**: (a) Satellite-derived SST (°C), photosynthetically active radiation (Einstein m$^{-2}$ d$^{-1}$), chlorophyll-*a* concentration (mg m$^{-3}$) and particulate inorganic carbon (PIC) concentration (mol m$^{-3}$) for the period November 2001 to September 2002. It is important to note that satellite PIC concentration estimates have been reported to be biased for high latitudes systems of the Southern Ocean where the satellite algorithm is thought to produce overestimates (Balch et al., 2016; Trull et al., 2017). Therefore PIC data presented here should be looked with caution. (b) Temporal variability of the total mass, calcium carbonate (CaCO$_3$) and particulate organic carbon (POC) the < 1mm fraction at 2000 and 3700 m water depth from November 2001 through to November 2002 at the 61°S site (Rigual-Hernández et al., 2015a). Grey strips represent summer.



## 3. Results

### 3.1 Seasonal dynamics of coccolith export fluxes

Coccolith fluxes showed a pronounced seasonal pattern at both sediment trap depths, roughly following the chlorophyll-*a* dynamics in the surface layer with maximum fluxes during the austral summer and minima during winter (Fig. 4 and 5). The summer coccolith particle bloom exhibited a bimodal distribution with a major peak registered in early January ($2.2 \ 10^9$ coccoliths m$^{-2}$ d$^{-1}$ at 2000 m) and a secondary maximum recorded in mid-February ($9.8 \ 10^8$ coccoliths m$^{-2}$ d$^{-1}$). Coccolith flux was low in autumn and winter (down to $7.5 \ 10^7$ coccoliths m$^{-2}$ d$^{-1}$). Coccolith fluxes in the deeper trap (3700 m) followed a similar pattern to that in the 2000 m trap with a delay of about one sampling interval.

The fluxes of all biogeochemical components were closely correlated (Table 2 in Rigual-Hernández et al., 2015a). Coccolith fluxes at both traps were broadly in line with biogenic particle fluxes estimated by Rigual-Hernández et al. (2015a) showing strongest correlations with Biogenic silica ($R^2 = 0.86$ at 2000 m and $R^2 = 0.71$ at 3700 m), followed by PIC ($R^2 = 0.62$ at 2000 m and $R^2 = 0.47$ at 3700 m) and POC ($R^2 = 0.56$ at 2000 m and $R^2 = 0.41$ at 3700 m).





Coccolithophore sinking assemblages at the 61°S site were nearly monospecific,
with an overwhelming dominance of *E. huxleyi* that represented >99% of the annual
coccolith sinking assemblage at both trap depths. Background concentrations of the
species *Calcidiscus leptoporus*, *Gephyrocapsa* spp. and *Helicosphaera* spp. were also
registered, together representing 0.6% and 0.3% at 2000 and 3700 m, respectively, of the

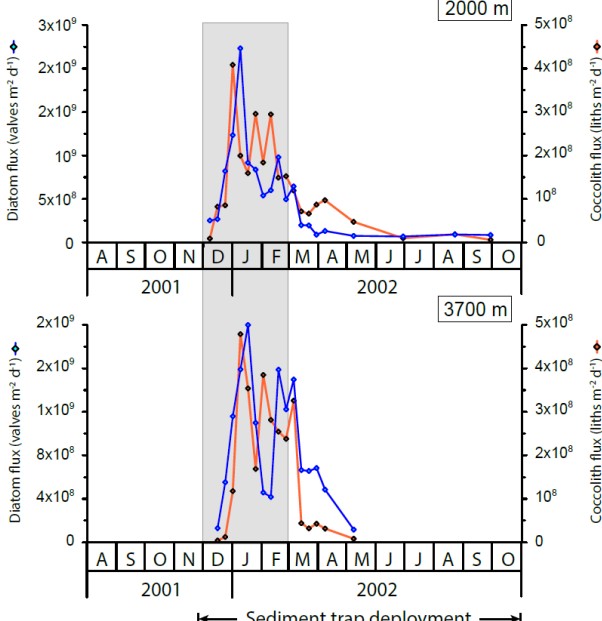

total annual coccolith fluxes (Table 1). The numbers of coccospheres found in the samples
were negligible in both sediment trap records.
**Figure 5:** Seasonal variation of total coccolith and diatom valve flux at the 2000 and 3700
m sediment traps at the 61°S site. Grey strips represent summer.

**3.2 SEM analyses**

*Emiliania huxleyi* coccoliths correspond to morphotype B/C, having proximal
shields slightly wider than the distal ones and with a central area usually filled by several
(usually 5 to 11) flat, wide and thin tiles (see Plate 1, image a). Several coccoliths present
distal shields partially missing, may be due to the slender and delicate structure of the
laths. Distal shield measures ranged between 2 to 4,35 µm in the samples recovered from
the 2000 m sediment trap. It is conspicuous that most of the coccoliths display a
morphology which is compatible with a secondary recrystallization. Small spherules like
recrystallizations are present on these coccoliths, especially on the laths (Plate 1, images
c-f). However, some coccoliths, mostly from cup 10 (February) have no spherules




covering them (Plate 1, images a and b). These coccoliths present very thin slender laths
(usually from 20 to 26) and wider central areas than the coccoliths having spherules.

**3.3 Coccolith weight and length changes**
Average coccolith weight at both sediment trap depths exhibited a clear seasonal
pattern with high values (2.3 and 2.1 pg/coccolith at 2000 m and 3700 m, respectively) at
the onset of the coccolithophorid bloom in early spring, followed by a pronounced
decrease (down to 1.6 and 1.9 pg at 2000 m and 3700 m, respectively) in approximately
late January – early February. Average coccolith weight followed a gradual increasing
trend from approximately mid-February into winter, reaching values up to 2.7 pg in
August 2002 at 2000 m and up to 2.43 in May at 3700 m, respectively. Average annual
coccolith weight at the 61°S traps was 2.11 ± 0.96 and 2.13 ± 0.90 pg at 2000 m and 3700
m, respectively. The annual amplitude of coccolith weight was approximately 1 pg at
2000 m and and 0.5 pg at 3700 m. The lower annual amplitude exhibited by the coccolith
assemblages captured at the 3700 trap is attributed to the lower sampling duration at that
depth over the winter season.
Mean coccolith length was greatest in early spring 2001 (3.1 and 3.2 µm at 2000
and 3700 m, respectively), followed by a decrease in early summer (down to 2.8 and 2.9
µm at 2000 and 3700 m, respectively) (Fig. 6). From late February coccolith length
increased again reaching the highest values of the record in winter 2002 (up to 3.2 and
3.3 µm at 2000 and 3700 m, respectively).
Seasonal variations of coccolith length and weight exhibited a strong correlation
at both depths ($R^2 = 0.84$, n = 20 at 2000 m; $R^2 = 0.61$, n = 16 at 3700m), indicating a
clear, dependable relationship between the two variables.



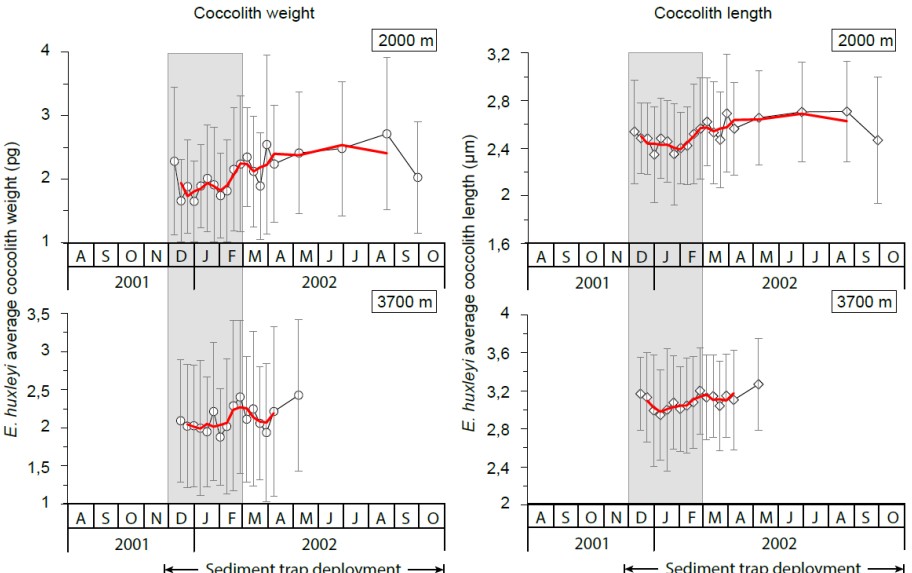


**Figure 6:** Mean and standard deviation of coccolith weight and length over the sediment
trap deployment period at 2000 m and 3700 m at the 61°S site. The red solid line
represents a 3-point running average. Grey strips represent summer.


**4. Discussion**
**4.1 Origin, magnitude and composition of the coccolithophores**

Since there is a current debate about the potential expansion of *E. huxleyi*
populations south of the Polar Front during recent decades (Cubillos et al., 2007; Winter
et al., 2014), it is important to evaluate the likely origins of the sinking coccolith
assemblages collected at station 61°S. This assessment is particularly needed in the case
of deep–moored, sediment-trap experiment because the source area of the particles
collected by the traps can be as wide as hundreds of square kilometres (Buesseler et al.,
2007).
Several lines of evidence strongly suggest that the coccolithophore fluxes
registered by the traps were produced in waters of the Antarctic Zone. Firstly, the mooring
was deployed in a quiescent area of the AZ-S (Trull et al., 2001b), between the stronger
flows associated with the southern branch of the PF and the SACCF (Fig. 1). The
relatively weak currents around the sediment trap location greatly reduce the area of likely
origins of the particles intercepted by the traps, i.e. the statistical funnel (Siegel and



Deuser, 1997; Siegel et al., 2008). Moreover, the large magnitude of the coccolith export
fluxes at both depths, plus the long duration of the "coccolith particle bloom" (about 3
months), rule out the likelihood of a transient lateral transport event (e.g., transport by
mesoscale eddies) of a coccolithophore bloom produced in more northerly latitudes.
Lastly, the composition of the biogeochemical fluxes and diatom assemblages collected
by the traps are characteristic of AZ waters (Rigual-Hernández et al., 2015a), further
supporting the idea that  the coccolithophores captured by the traps  were produced close
to the site. All this clearly indicates that in 2001 $E.$ $huxleyi$ was an established member of
the phytoplankton communities of the Antarctic Zone to the south of Australia.

The annual coccolith export to the deep ocean at the 61°S site (1.03 x $10^{11}$

coccoliths  m$^{-2}$ yr$^{-1}$) is one sixth that registered by Wilks et al. (2017) (6.5 x $10^{11}$ coccolith
m$^{-2}$ yr$^{-1}$ ) in the SAZ waters (station 47°S; Fig. 1) north of the study site. The lower
abundance of coccolithophores at the 61°S site is most likely due to the negative effects
of low temperature and low light levels on coccolithophore growth (Paasche, 2002; Boyd
et al., 2010), but important also is the competitive advantage of diatoms over
coccolithophores in the silicate-rich waters of the AZ. The lower coccolithophore
production in the AZ is also reflected in the lower carbonate export at this site, i.e. 6 g m$^{-2}$
$^{2}$ y$^{-1}$ versus 10-13 g m$^{-2}$ y$^{-1}$ at the 47°S site (Rigual-Hernández et al., 2015b; Wilks et al.,
2017). The non-proportional latitudinal change in coccolith and carbonate fluxes (i.e.
sixfold versus twofold changes, respectively) is most likely due to variations in the
contribution of heterotrophic calcifiers (i.e. foraminifers and pteropods) to total carbonate
export. There are also differences in the carbonate content per coccolith of the
coccolithophore species and the morphotypes of $E.$ $huxleyi$ dwelling in each zonal system.
Indeed, mean coccolith weight can vary up to two orders of magnitude between small
species such as $E.$ $huxleyi$ (2-3.5 pg) and large and heavily calcified taxa such as
$Coccolithus$ $pelagicus$ (~150 pg) (Giraudeau and Beaufort, 2007). Intraspecific size
variability is also common in most coccolithophore species, mainly due to growth
variations driven by different environmental factors and by genotypic variability (e.g.
Knappertsbusch et al., 1997; Poulton et al., 2011).

Recognizing the significant genetical variability they found between Southern

Ocean populations of morphotypes A and B/C, Cook et al. (2011) classified these
morphotypes as $E.$ $huxleyi$ var. $huxleyi$ and $E.$ $huxleyi$ var. $aurorae$, respectively. Since
only morphotype B/C had been reported at and south of the Antarctic Polar Front, Cook
et al. (2013) concluded that the rapid drop in water temperature occurring at the Antarctic



Polar Front may act as an open-ocean barrier to gene flow between these the two Southern
Ocean *E. huxleyi* morphotypes/varieties. The monospecific coccolith assemblages of *E.*
*huxleyi* morphotype B/C collected by the 61°S site traps (Plate 1) are consistent with those
studies and supports the idea that the physiological differences in light-harvesting
pigments of morphotype B/C compared to other *E. huxleyi* varieties (Cook et al., 2011)
may represent a critical ecological advantage in the cold and low-light waters of the AZ
south of Australia.

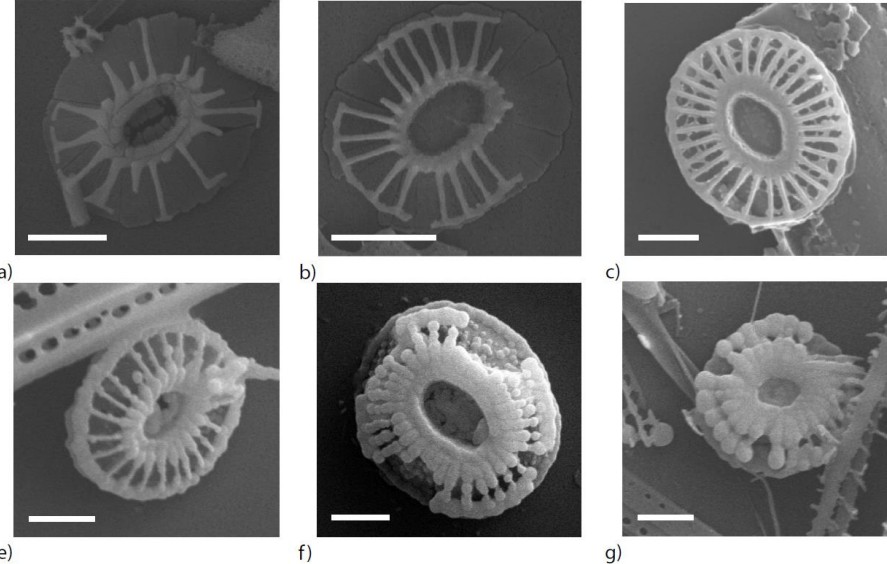

**Plate 1:** SEM photos showcasing the different morphologies of *Emiliania huxleyi*
morphotype B/C coccoliths found in the sediment traps of the 61°S site. Scale bars =1
μm.

**4.2 Seasonal dynamics of the calcareous and siliceous phytoplankton fluxes**
The eight-day sampling resolution during spring and summer enabled us to
monitor the detailed temporal dynamics of phytoplankton fluxes at the 61°S site.
Comparison of satellite-derived PIC and Chl-*a* concentrations for the study region with
coccolith fluxes registered by the sediment trap suggests a time lag of about two months
between their surface maxima and peak coccolith fluxes registered by the shallower trap
(Fig. 4). Therefore, the growth phase of the *E. huxleyi* bloom probably took place between



October and December 2001, a period characterized by very low SSTs (0.1-0.9 °C). It
was before development of any significant stratification in the upper water column (Fig.
2b and 4a). These observations indicate that the very cold temperatures (near 0°C) and
strong mixing of the water column in the Antarctic waters during spring are not an
impediment for the development of an *E. huxleyi* bloom.

The onset of seasonal increase in coccolithophore arrivals in the traps occurred at

the same time as that of diatoms, suggesting a rapid response of both phytoplankton
groups to enhanced light levels. Although both coccolith and diatom fluxes exhibited a
pronounced and nearly parallel increase throughout December (Fig. 5), coccolith fluxes
peaked one week later than those of diatoms. A similar succession was observed in late
summer, when coccoliths displayed a secondary flux maximum, one sampling interval
later (8 days) than that of diatoms (Fig. 5). These observations agree with the bloom-
dynamics scheme proposed by Barber and Hiscock (2006) (the so-called coexistence
theory), in that neither phytoplankton group seems to outcompete the other during the
development of the bloom. Interestingly, diatoms seem to decline  earlier than
coccolithophores, a feature often (but not always) observed in other parts of the world
ocean (e.g. Margalef, 1978; Holligan et al., 1983; Lochte et al., 1993; Sieracki et al., 1993;
Thunell et al., 1996; Balch, 2004). Indeed, a recent study of the phenological
characteristics of coccolithophore blooms by Hopkins et al. (2015) concluded that they
often follow those of diatoms in many regions,  the sequencing driven by increasing
stabilization and/or nutrient depletion (mainly silicate and/or Fe, and possibly also
favoured by associated increase of carbonate saturation; Merico et al, 2004)  of the surface
layer.

Lack of nutrient and mixed-layer-depth measurements during the sediment trap

deployment precludes us from establishing robust links between changes in physical and
chemical parameters in the upper water column and the observed phytoplankton
succession. Nonetheless, some shipboard observations of mixed-layer properties from
years previous to the sediment trap deployment (Fig. 2; Trull et al., 2001b) can provide
some insight about the mechanisms driving the phytoplankton succession. Macronutrient
measurements indicate that, although considerable nutrient draw-down often occurs by
mid-summer, the AZ-S waters never reach potentially limiting concentrations (i.e. below
10 µM) of silicate, nitrate or phosphate (Fig. 2a; Trull et al., 2001b). Thus, macronutrient
limitation was not a likely driver of the observed phytoplankton succession at the 61°S
site traps. Iron levels in the AZ-S, on the other hand, are low year-round (0.1-0.2 nM;

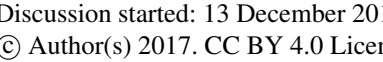


Boyd et al., 2000b; Sohrin et al., 2000) and exhibit clear seasonality in the AZ (Tagliabue
et al., 2014). So, iron availability does represent a potential driver for the observed
phytoplankton succession. Indeed, laboratory experiments have shown that *E. huxleyi* has
lower minimum Fe requirements for growth than oceanic diatoms (Brand et al., 1983;
Muggli and Harrison, 1997). This physiology likely provides an ecological advantage
over diatoms in the later stages of the spring-summer bloom, when most iron has been
stripped from the mixed layer.
In regard to the mechanism underlying the second diatom-coccolith succession
observed at both depths in February (Fig. 5), it is possible that a vertical mixing event –
as frequently reported in the AZ (e.g. Brzezinski et al., 2001) – supplied waters rich in
iron and macronutrients to the euphotic zone, resetting the phytoplankton succession.
Alternatively, the part of the *Emiliania huxleyi* populations accumulated at or just above
the nutricline may have increased using the iron moved by diapycnal diffusion through
the pycnocline (Tagliabue et al., 2014). Their deposition in February could have been
triggered by a drop of the light levels (Fig. 4). This second hypothesis is also consistent
with the following observations: (1) the presence of a sub-surface chlorophyll-a
maximum in the study region during spring and summer (Parslow et al., 2001; Trull et
al., 2001b); (2) reports of high *E. huxleyi* cell accumulations associated with the nutricline
in other settings of the world ocean (Beaufort et al., 2008; Henderiks et al., 2012) and (3)
peak annual sedimentation in late February of the diatom *Thalassiothrix antarctica*
(Rigual-Hernández et al., 2015a), a typical component of the "shade flora" (Kemp et al.,
2000; Quéguiner, 2013). Further sampling and taxonomic analysis of the vertical
distributions of phytoplankton in the AZ south of Australia are required to asses these
hypotheses.

**4.3 Seasonal variability in coccolith calcification**
Two main factors have been proposed as driving seasonal changes in coccolith
weights of *E. huxleyi*: a seasonal shift in the dominant morphotypes/ecotypes — each
with a different degrees of calcification (Poulton et al., 2011) — and the physiological
response of a given morphotype to the seasonal variation of environmental parameters
(e.g. Smith et al., 2012; Meier et al., 2014). SEM analysis of the 61°S sediment trap
samples revealed that only morphotype B/C, *sensu* Young et al. (2003), thrives in the AZ-
S waters south of Tasmania. That is consistent with a report by Cubillos et al. (2007) of
dominance of B/C south 50°S. Therefore, a seasonal shift in the dominant morphotype



can be ruled out in respect to changing coccolith weight. The observed decrease in
coccolith weight could have been caused by a change in coccolith calcification or
reduction in coccolith dimensions. Young and Ziveri (2000) showed that coccolith weight
is approximately linearly correlated to the cube of coccolith length. Applying that, the
decrease in length by 7.5% (a reduction to 92.5%) observed from the pre-bloom to the
summer bloom in the 2000 m traps (i.e. difference in minimum coccolith lengths in cups
5 and 8) corresponds to a coccolith weight loss of 21% ($0.925^3 \approx 0.79$). That is similar to
the observed weight reduction in the 2000 m trap between the pre-bloom and summer
bloom coccolith assemblages (16.2 - 27.6%, respectively Fig. 6). When the linear
correlation between coccolith length and weight proposed by Young and Ziveri (2000) is
also applied to the 3700 m trap coccoliths, the predicted reduction of coccolith weight
between the pre-bloom and bloom assemblages is 12%. That is again very similar to the
reduction in coccolith weight observed in the *E. huxleyi* coccoliths intercepted by the
3700 trap (10%). It is strongly suggested that the seasonal changes in coccolith weight at
the 61°S site were mainly driven by changes in coccolith length and were not due to
significant changes in their degrees of calcification.
Laboratory, mesocosm and field studies have shown that multiple environmental
factors including irradiance, temperature, macronutrient concentrations and iron
availability affect coccolith formation by *E. huxleyi* cells (e.g. Paasche, 2002; Zondervan,
2007; Langer and Benner, 2009; Feng et al., 2017). Since calcification in *E. huxleyi* is a
light-dependant process (Paasche, 1999, 2002), the observed decrease in coccolith weight
during summer in both traps was somewhat unexpected. Some field experiments have
shown that calcification in coccolithophores can occur at low light levels, or even in the
absence of light (e.g. van der Wal et al., 1994). However, it is often reduced compared to
that at higher irradiance levels (Zondervan, 2007).
In terms of temperature effects, Feng et al. (2017) showed that optimal
temperature for calcification on *E. huxleyi* cultures (morphotype A, strain NIWA1108)
was ~20°C, while temperatures below 10°C resulted in a dramatic reduction of
calcification rates and severe malformations of coccoliths, such as incomplete distal
shield elements. Although *E. huxleyi* morphotype B/C found at the 61°S site likely
represents an ecotype more tolerant to low temperatures than morphotype A (Cubillos et
al., 2007; Cook et al., 2013), the frequent variations in the structure of the coccoliths (e.g.
incomplete distal shield elements; Plate 1) captured by the traps suggest some degree of
low-temperature stress at the 61°S site. Despite the important role of temperature in



coccolithophore growth (Paasche, 2002), enhanced summer SSTs may lead to an increase
in coccolith calcification, a  response opposite to that observed at both traps. Therefore,
it is unlikely that seasonal SST variations at the 61°S are behind the observed variability
in coccolithophore weight.
In regard to the possible impact of macronutrient concentrations on coccolith
weight, both nitrate and phosphate are known to have a pronounced effect on coccolith
calcite content and morphology (Zondervan, 2007). However, as mentioned previously,
none of these macronutrients reach limiting concentrations throughout the annual cycle
in the AZ (Fig. 2; Trull et al., 2001). and, therefore, their influence in the calcification of
coccolithophores is likely to be low or negligible.
On the other hand, low iron levels have been reported to have a pronounced
negative effect on $CaCO_3$ production by *E. huxleyi* cells (Schulz et al., 2004), so it
represents a candidate driver of seasonal changes in coccolith weight. During winter, deep
water mixing re-stocks the mixed layer with iron (Tagliabue et al., 2014). As soon as light
levels become sufficient for photosynthesis in early spring, phytoplankton rapidly
develops under non-limiting concentrations of macro- and micronutrients. These
favourable conditions for coccolithophore growth could explain the heavier and larger
coccoliths registered in early December (Fig. 6). As the phytoplankton bloom develops,
the dissolved iron stock is rapidly depleted in the photic zone possibly resulting in a size
and weight reduction of coccoliths of the already substantial *E. huxleyi* populations. From
late summer throughout autumn, some recycling of iron in the upper water column by
increasing summer populations of zooplankton feeding on the bloom (Tagliabue et al.,
2014), coupled with increasing light levels and the continued shallowing of the mixed
layer, would allow coccolithophores to produce again longer and heavier coccoliths (Fig.

6).

Iron-limitation, therefore, represents the most likely environmental driving factor
for the seasonal variability in coccolith weight and length of *E. huxleyi* assemblages at
the 61°S site. However, we note again that the absence accompanying *in situ*
measurements of chemical and physical parameters of the water column, means that
control of coccolith weight by varying iron availability in the AZ remains an hypothesis
needing validationby future studies.

**4.4 Effects of calcite dissolution on the sinking coccolith assemblages**





The similar average annual coccolith weight registered at both traps (2.5 pg at
2000 m to 2.6 pg at 3700 m) indicates that negligible coccolith dissolution occurs at meso-
and bathypelagic depths in the AZ south of Australia. That is despite the fact that
coccolith sinking assemblages captured by the deeper trap were exposed to potentially
intense dissolution after crossing the CSH (located at 3000 m in the study region; Fig. 2).
The similar coccolith values observed at both depths can be attributed to the formation of
algal and faecal aggregates in the mixed layer that include fine mineral particles (Passow
and De La Rocha, 2006) and provide protection against dissolution. They also facilitate
rapid transport of the coccoliths down through the water column. The aggregate-
formation hypothesis is supported by the findings of Closset et al. (2015) who estimated
that sinking rates at the 61°S site were, at least 213 m d$^{-1}$ during the productive period, a
value consistent with the sinking rates of algal and/or faecal aggregates (Turner, 2002;
Turner, 2015).
Despite not finding increased dissolution with water depth between 2000 and 3700
m, it is possible that coccoliths experienced some carbonate dissolution before reaching
the traps. Milliman et al. (1999) suggested that the same biological processes that
facilitate aggregate formation and flocculation, such as ingestion, digestion and egestion
by grazers, may be responsible for significant carbonate dissolution at epipelagic depths
(i.e. depths shallower than 800-1000 m. Indeed, the negligible amounts of coccospheres
found in both traps, together with the high sinking velocities, suggest that grazing could
have been an important influence on export. That supported by findings of Ebersbach et
al. (2011) in the PFZ north of 61°S. They documented that an important fraction of the
particles sinks from the mixed layer as faecal aggregates. However, the available data are
insufficient to evaluate the impact of carbonate dissolution in the upper water column.

**4.5 Calcium carbonate content of *Emiliania huxleyi* coccoliths**
A broad range of calcite contents for *E. huxleyi* coccoliths (1.4 - 7.0 pg) has been
proposed in the literature (e.g. Young and Ziveri, 2000; Beaufort, 2005; Holligan et al.,
2010; Poulton et al., 2011). The differences in these estimates are most likely due to
variability in the amount of coccolith calcite between morphotypes and to the varied
methodological biases associated with the three main approaches for estimating coccolith
mass: morphometrics, regression and birefringence. Since *E. huxleyi* morphotype B/C is
more weakly calcified than other morphotypes (Young and Ziveri, 2000) and considered
to be geographically restricted to the Southern Ocean (Cubillos et al., 2007; Cook et al.,

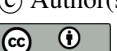



2013) we limit the comparison of our results to studies conducted only in the Southern
Ocean.

Average annual coccolith quotas at both trap depths at the 61°S site (2.11 ± 0.96

and 2.13 ± 0.90 pg per coccolith at 2000 m and 3700 m, respectively) are almost identical
to that estimated by Holligan et al. (2010) (2.20 ± 0.60 pg ; morphotype B/C) in the Scotia
Sea using a regression line between the number of coccoliths against PIC. Moreover, our
estimates are slightly higher, but with a considerable overlap in the ranges of coccolith
weight, than those estimated by Poulton et al (2011) for the *E. huxleyi* morphotype B/C
populations found in Patagonian shelf waters (1.40 ± 0.6 pg). The greater standard
deviation of our data is most likely due to the time periods compared. While the average
coccolith weight estimated for our traps reflects an integration of the annual variability in
coccolith weight, the shipboard observations by Poulton et al. (2011) provide a snapshot
of the summer coccolithophore populations, that likely exhibit lower coccolith size and,
thus, variability.

Because our coccolith weight estimates are similar to those of Poulton et al. (2011)

and Holligan et al. (2010), we can estimate the fractional contribution of coccolithophores
to total carbonate production in the AZ south of Australia. Coccolithophores account for
approximately 2-5% of the annual deep-ocean $CaCO_3$ fluxes at mesopelagic depths at the
61°S site. The contribution of coccolithophores to the annual $CaCO_3$ budget in the AZ
south of Australia is similar to the estimate by Salter et al. (2014) for the macronutrient-
rich, but iron deficient M6 site in the Indian sector of the AZ (12%) and remarkably lower
than an estimate for the iron-fertilized station A3 over the central Kerguelen Plateau
(85%; Rembauville et al., 2016). Due to the different methodologies for estimating
coccolithophore contributions to carbonate production, comparison of our results with
these other studies should be treated with caution. While only whole coccoliths were
counted for our calculation, therefore providing a conservative estimate, Salter et al.
(2014) and Rembauville et al. (2016) estimated the weight of the < 20 μm fraction using
inductively coupled plasma-atomic emission spectrometry. That approach often results in
overestimates of the coccolith contribution to bulk carbonate content. There can be non-
negligible contributions of non-coccolith fragments to the fine fraction (Giraudeau and
Beaufort, 2007). Despite the biases associated with both methodologies, the general trend
appears clear: the fractional contributions of coccolithophores to bulk carbonate export
are lower in the iron-limited waters of the AZ compared to those in naturally iron-
fertilized settings of the Southern Ocean. These findings underscore the secondary role





of this phytoplankton group in the biological carbon pumps (both the in organic carbon
and carbonate counter pumps) south of the PF where non-calcifying phytoplankton -
mainly diatoms and *Phaeocystis* – largely control the biologically-mediated $CO_2$
exchange between the ocean and the atmosphere.

**Conclusions**

Analysis of the sediment trap materials captured at the 61°S site allowed for the
characterization and quantification of coccolith assemblages in Australian sector of the
Antarctic Zone, providing a baseline of the state of coccolithophore populations in this
region against which future changes can be assessed. More specifically, our study has
shown the following:
• Coccolithophores were a consistent member of the phytoplankton communities of the
Antarctic Zone south of Australia in year 2001. Coccolithophore assemblages in this
region are monospecific being composed almost entirely of *Emiliania huxleyi*
morphotype B/C. This observation supports the hypothesis that the physiological
differences in light-harvesting pigments of morphotype B/C (or *E. huxleyi* var.
*aurorae*), compared to other Southern Ocean *E. huxleyi* varieties (Cook et al., 2011),
may represent an ecological advantage in the cold, low-light and iron-limited
environment of the Antarctic Zone.
• The onset of the coccolithophore productive period took place at the same time as that
of diatoms, indicating that neither phytoplankton group outcompetes the other during
the development of the bloom. We speculate that the diatom-coccolithophore
succession observed during the peak phase of the productive period could result from
the lower minimum iron requirements for growth of *E. huxleyi*, a feature that may
confer a competitive advantage over diatoms.
• A decrease in coccolith weight and size during the summer months was observed at
both sediment trap depths. After assessing the potential influence of several
environmental parameters, increasing iron limitation seems to be the most likely
candidate to drive this change. This hypothesis, however, will need to be validated in
future field and laboratory culture experiments with morphotype B/C.
• The similar weight of *E. huxleyi* coccolith assemblages captured by the 2000 and 3700
m sediment traps indicates that negligible coccolith dissolution occurs during transit



through meso- and bathypelagic depths in the study region. This is most likely due to
a rapid transport of the coccoliths in algal and/or faecal aggregates.
• Coccolith weight values calculated for both sediment trap records using a
birefringence-based approach were similar to previous estimates of *E. huxleyi*
morphotype B/C in other Southern Ocean settings using regression and morphometric
methods (Holligan et al., 2010; Poulton et al., 2011, respectively).
• Coccolithophore fluxes at the 61°S site account for only 2-5% of the annual deep-
ocean $CaCO_3$ fluxes, suggesting that heterotrophic calcifiers must represent the main
biogenic carbonate producer in the AZ south of Australia.

## 734   Acknowledgments

Sediment trap deployments and sample processing were carried out by ACE CRC staff
(including Stephen Bray and Diana Davies) with support from the Australian
Commonwealth Cooperative Research Centres Program and Australian Antarctic
Division (via AAS Awards 1156 and 2256 to T. W. Trull). We thank Charlie B. Miller
for English corrections and comments on the manuscript.

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
