# Peer review of "Coccolithophore populations and their contribution to carbonate export during an"

_Biogeosciences, 2017_

## Referee Comment (RC1) · Anonymous Referee #1 · 12 Jan 2018

The manuscript documents an annual record of coccolithophore production and coccolith weights/lengths at a Southern Ocean site. The topic is worthy of publication in biogeosciences, and the results will be of potential interest to the wider scientific community. In general, the manuscript is well written and illustrated, and does not contain any major flaws. However, see below for minor points.

Scientific points Line 38: " … coccolith assemblages experienced weight and length reduction .." !? It is not the assemblages that have reduced weight and length, it is the coccoliths.

Line 89: the Southern Ocean is a small area ? – 25% of the global area looks quite significant to me

Line 148: why is this section in the methods ? move regional setting and oceanography to the introduction

Line 235: why unfiltered ? I am not a specialist on sediment traps but it seems odd to use unfiltered seawater. Is there not a risk of contamination ?

Lines 357 and 345: "coccolith particle bloom" – since coccoliths are inanimate (just pieces of calcium carbonate) I think the word 'bloom' is inappropriate here – use 'The summer coccolith flux exhibited . . .'

Line 548: you mention two factors that possibly explain the changes in calcification. Calcification (i.e. overgrowth) tends to increase with depth in the photic zone, at least in some areas of the world. So in winter it may be that the coccolithophores are sitting deeper and therefore have more calcified coccoliths than in the summer when they are closer to the surface and therefore with lightly calcified coccoliths. Of course this difference in surface vs deeper photic could be related to various parameters (light, nutrients, temperature). Do you have data/images of coccoliths from different photic depths ? In Plate 1 you show lightly and heavily calcified coccoliths from the traps - but how do they relate to the surface oceans ?

Minor points for correction/consideration Title and elsewhere: Just a query. Is the use of Australian Sector OK ? Naming the sectors after the oceans, like the Atlantic Sector, Pacific Sector and Indian Ocean Sector is fine, but I wonder whether using country names (for sectors and territories) is considered to be geopolitical.

Line 30: Don't mix z and s verbs. For example, here you use 'characterized' and on line 151 'summarized', but on line 35 you use 'analysed' and on line 135 'fertilisation'. Furthermore, on line 236 you use 'programmed' and on line 349 'grey'. You need to be consistent, and choose between British English and US English. It looks like you are

favouring the former.

Line 45: coccolithophorid vs coccolithophore. Be consistent, and choose one.

Line 62: " ..some species (but not all) of coccolithophore .." – please change to 'some species of (but not all) coccolithophores ..'

Lines 71-75: needs to be rewritten, as it doesn't make sense

Line 79: one bracket is missing

Line 102: dominantly present -> dominate

Line 115: spares -> sparse

Line 149: ".at the north . . . at the south .." -> ' ..in the north . . . in the south ..'

Line 203: CO2 rich -> CO2-rich

Line 253: "After settling 12 hours .." -> 'After settling for 12 hours ..' Line 271: by using -> using

Line 273: " ..to the winter .." -> ' ..to winter ..'

Line 285: Scanning Electron Microscopy -> scanning electron microscope (SEM)

Line 298: "..using a with a Nikon .." -> ' . . .using a Nikon ..'

Line 346: " ..should be looked with caution .." -> 'should be viewed with caution'

Line 358 (and elsewhere) : you need to insert x (times) between the number and the power. For example, 2.2 10 -> 2.2. x 10

Line 365: Biogenic -> biogenic

Line 370: of the species Calcidiscus -> of Calcidiscus

Figure 5 (and elsewhere): I realise that 'liths' is in common use in presentations, but it is not an official term. Better to use coccoliths.

Line 382: tiles? I think you mean 'elements'

Line 384: "Distal shield measures ranged between 2 and 4,35 .." -> 'Distal shield ranges from 2.0-4.35 ..' [use decimal point not comma]

Line 424: here you use station 62 S, and before 62 S site – perhaps be consistent in usage

Line 463: genetical -> genetic

Line 550: degrees of calcification -> degree of calcification

Line 555: B/C south 50°S -> B/C south of 50°S

Line 576: light-dependant -> light-dependent

Line 617: absence accompanying in situ -> absence of in situ

Line 643: That supported -> That is supported

Lines 741-759: delete, as the same as later references

Lines 773 and 816 (and elsewhere): Deep Sea -> Deep-Sea Line 860: emiliania huxleyi (haptophyta) Âź, Journal of phycology, -> Emiliania huxleyi 860 (haptophyta) Âź, Journal of Phycology, [why is there a superscript 1 at the end of the title ?

Lines 864, 867 and 884 ( and elsewhere): italicize the species name

---

## Referee Comment (RC2) · Anonymous Referee #2 · 22 Jan 2018

**REVIEWERS COMMENTS**

The calcification of coccolithophores in the high latitude regions is a growing concern as it will have large influence on ocean biogeochemistry and thus climate. The data provides new information on coccolithophore response to varying environmental conditions at Antarctic Zone using sediment traps samples. However, the information provided here can be expressed in a much better way. Sediment trap data from Southern Ocean is difficult to obtain and is therefore a welcome addition to knowledge and needs to be published. Here are some major and minor comments which need to be incorporated in the manuscript.

**Major comments/changes needed to be done:**

1. Authors documented only abundance of coccoliths of E. huxleyi B/C morphotypes in the traps. It is also mentioned that occurrence of other coccolithophores were also documented. Though other coccoliths are in low abundance, it represents changes in the upper oceanic conditions. It is also important to plot a graph of other coccolithophores and discuss what their assemblage indicates.

2. The overcalcification of E. huxleyi is documented by few researchers in past few years. But not many papers are published on this. Authors have documented overcalcification on coccoliths retrieved from sediment traps. I assume intact coccospheres are also documented in the both the sediment traps. In this case, whether authors documented overcalcification on coccospheres of E. huxleyi? If any information is available on living coccolithophores in this region, it should be included. It is important to document the overcalcification on E. huxleyi is a natural process and not a part of secondary calcification. So, if extant coccolithophores data is available at/around study site then it should be presented.

3. Authors often compared diatom assemblage with coccoliths. Plotting a graph of total diatom assemblage vs coccoliths abundance will be useful for understanding changes in the AZ region.

4. A recent study indicates polysaccharides are also responsible for overcalcification of coccolithophres. Authors need to discuss outcomes in more detail and should be cited with recent references.

5. Authors should be consistent in framing sentences. Some sentences are too large, some are too small. Be consistent in using AZ vs AZ-S, Fe vs. iron, E. huxleyi, vs Emiliania huxleyi etc.

6. Since, both sediment traps are located in the 61degS and there is no comparison done with other sediment traps showed in the Fig. 1, it is not necessary to mention "at 61S sediment trap everywhere"

7. Authors should crosscheck references very carefully. Many references listed in the text are not reflected in the reference list. Similarly, many references listed in the Reference list are not

mentioned in the text. Genus and species name should appear properly and in italics. For ex. Line 860- emiliania huxleyi.

Hagino et al. 2011 reference- written in caps

**Minor comments:**

Line 14-30: The information provided in the abstract can be shifted to the introduction. The abstract should start from Line 31.

Line 32: In the deep ocean >>> at the Antarctic Zone

Line 33: ~2000>>> 2000

Line 33: and 3700>>> and 3700 respectively

Line 37: Emiliania huxleyi morphotype B/C>>>> E. huxleyi morphotype B/C

Line 38: coccolith assemblage experienced weight and length reduction>>> coccoliths experienced weight and length reduction

Line 39: during the summer months>>> during summer months (December-March?)

Line 40: at both sediment trap depths>>> at both sediment traps

Line 41: in other southern ocean settings>>>> which settings?

Line 43-46: Apart from first record, significant outcomes of the study needs to be highlighted here.

Line 71-75: May not required in the introduction

Line 76: Coccolithophores also has the potential >>>> coccolithophores has potential

Line 89: 25% of ocean area is not small.

Line 107: Trull et al., 2017 is not mentioned in reference list

Line 109: Cubilos et al. 2008 or 2007? Cubilos et al. 2008 is not mentioned in the reference list

Line 111: Freeman and Lovenduski (2015) not mentioned in the reference list

Line 116-122 and elsewhere in the introduction: very large sentence. Authors should be consistent in framing sentences. Such long sentences to be avoided in the introduction

Line 123: Trull et al., 2017 missing in the reference list

Line 138: inferred from one-year record>>>> inferred from 10 month record

Line 139: SOIREE……. Elaborate when using short forms for the first time

Line 147: Regional setting and oceanography; and water carbonate chemistry, should be shifted in the introduction under a different sub-heading.

Line 151… SAACF>>>SACCF

Line151: (SAACF)>>>(SACCF) Fig.1

Line 155: upper water column with nutrients (add reference). Similarly for sentences between lines 155-160 (add references)

Line 155: Chlorophyll-a, vs/ Line 484 chl-a>>> use one style of writing

Line 160: ….. in algal biomass occurs within the mixed layer (add reference)

Line 164: Trull et al. 2001>>>> Trull et al. 2001a or Trull et al. 2001b or 2001c??

 Line 169: Large calanoid copepodites.>>> Copepods and copepodites are different. Copepodites are immature form of copepods. What authors are trying to say? "large Calanoid copepods" or "mainly Calanoid copepodites"??

Fig. 1. Since author has mentioned that sediment trap location was away from sea ice activity, authors should draw seasonal sea ice zone or winter sea ice limit for the year 2001-2002 in Fig. 1

Line 180: SACCF- Southern ACC Front>>>> SACCF- Southern Antarctic Circumpolar Current Front

Line 193: calcite saturation horizon (CSH)>>>> Calcite Saturation Horizon (CSH)

Line 194: CaCO3 compensation depth (CCD)>>>> CaCO3 Compensation Depth (CCD)

Fig. 2. Similar to Fig. 1 and Fig. 3, Fig. 2 should be plotted in the Ocean data view and figures should be readable. What does the dotted line indicate in Fig. 2b?

Fig 2 a. legend should contain surface macronutrient concentrations?

Line 200: (Fig. 3)>>> (Fig.3a) or (Fig. 3b)

Fig. 3. Mark Fig. 3a and Fig 3b; Mark frontal locations, put units for color scale bar, x axis etc. Also, include sampling dots if possible. Mark 1000m sediment trap in fig 3 in different color., as it is mentioned in line 226.

Line 210: Elaborate when appear for the first time

Line 211: Tanhua et al. 2008 is missing in the reference list

Line 122: Draw seasonal sea ice zone in Fig. 1

Line 225: for approximately one year>>>> for ten months

Line 227: highlight Fig. 3a and 3b

Line 231: be consistent in using depths. Either use "~2000" or "2000",,,, "3800 or 3700

Line 283: Emiliania huxleyi>>>> E. huxleyi

Line 235: why unfiltered seawater used? Won't it contaminate samples?

Line 285: Scanning Electron Microscopy>>>> Scanning Electron Microscope (SEM)

Line 286: decantation method outlined by>>>> method outlined by

Line 287: coated in Gold>>> coated with/using Gold

Line 288: please mentioned magnification range (for example 2000-7000x) used during analysis.

Line 288: Emiliania huxleyi>>>>E. huxleyi

Line 291: Emiliania huxleyi>>>>E. huxleyi

Line 293: Emiliania huxleyi>>>>E. huxleyi

293-294: Emiliania huxleyi coccoliths into morphotypes>>> Emiliania huxleyi into different morphotypes

Line 298: using a with a Nikon>>>> using a Nikon

Line 319: sea surface temperature (SST) already elaborated in line 153

Line 320: Sea Surface Temperature Analysis>>>> SST Analysis

Line 323: SST variations>>>> Sea Surface variations

Line 326: Photosynthetically active radiation (PAR)>>>> Photosynthetically Active Radiation (PAR)

Line 327: particulate inorganic carbon (PIC)>>>> Particulate Inorganic Carbon (PIC)

Fig. 4. Authors have mentioned March as a late summer months in the line 158. In this case, the shaded area should also include March

Line 341: particulate inorganic carbon (PIC)>>>> PIC

Line 342-346- It is important……………………….. Trull et al., 2017)>>> already mentioned before

Line 347: particulate organic carbon (POC)>>>> POC; Calcium Carbonate (CaCO3)>>>> CaCO3

Fig. 5. Please check scale bars. $2x10^9$ appeared twice on left side.; in the first figure only $10^9$ appeared. Is it $1x10^9$?. Put same scale in both figures. Mark Figure 5a and 5b.

Fig. 5. What is the reason diatom valve flux remained almost constant at 2000 and 3700 but there is an increase in diatom flux during February and March. Please explain.

Line 370-372: which Calcidiscus leptoporus species? Small or intermediate? Which time of sampling month these species are documented?

Line 383: distal shields partially missing, may be due to >>>> distal shield partially missing, due to

Line 384: 2 to 4,35>>>> 2 to 4.35

Line 394: (2.3 and 2.1 pg/coccolith)>>>> (2.3 ±….   and 2.1±….  pg/coccolith)

Lien 396: (down to 1.6 and 1.9 pg at 2000 and 3700m, respectively)>>>> (down to 1.6±…. and 1.9±…. pg/coccolith at 2000 and 3700m)

Line 399: Average annual coccolith weight at the 61S traps>>>>it is already mentioned that both traps are located at AZ-S, at 61S. just mentioned depths. Similarly at Line 433, at the 61S site>> study site. Similarly correct changes at line 446 and elsewhere

Line 405-406: what makes E. huxleyi coccolith change their lengths in early spring to summer discuss under discussion. Please refer Bollmann et al paper.

Line 410-412: If possible, plot graphs of correlation

Line 422: South of the Polar Front>>>> South of the PF

Line 422: include recent studies carried out in the Southern Ocean such as, Patil et al., 2017, Saavedra Pellitero et al, Malinverno et al.,

Line 426: Buesseler et al., 2007 reference is not in the reference list

Line 435: What author mean by "coccolith particle bloom"?

Line 450: use either AZ or AZ-S

Line 463: general variability they found between>>>> general variability found between

Line 463: please differentiate morphological differences between Morphotypes A, B/C, var. huxleyi and var. aurorae. E. hxuelyi morphotype C is usually less calcified morphotype of E. huxleyi usually found in the AZ. What are the probable reasons for absence of morphotype C in sediment trap samples?

Plate 1: I don't agree with all six images belonging to morphotype B/C. Plate 1b, e, looks like morphotype C. Plate 1. g is unrecognizable due to overcalcification. Authors can follow Young et al., 2003 atlas for differentiating E. huxleyi morphotypes.

Line 508: Silicate and/or Fe>>>> Silicate and/or iron

Line 522, line 535: Tagliabue et al., 2014. Year of publication missing in the reference list

Line 533: the part of the Emiliania huxleyi>>>> the part of the E. huxleyi

Line 598: Trull et al., 2001>>>>> Trull et al., 2001a or 2001b or 2001c?

Line 623: both traps (2.5 pg at 2000m to 2.6 pg at 3700)>>>>> both traps (2.5±…. pg at 2000m to 2.6±…. at 3700m)

Line 634: (Turner, 2002: Turner, 2015)>>>> (Turner, 2002; 2015)

Line 655: E. huxleyi morphotype B/C is more weakly calcified than other morphotypes>>>> I am not convinced with this. E. hulxleyi morphotype C is more weakly calcified than B/C. It can be written as >> E. huxleyi morhotype B/C is weakly calcified than A> if authors want to tell extent of calcification.

Line 676, 682: Salter et al. (2014) missing in the reference list

Line 679, 683: Rembauville et al. missing in the reference list

Line 698: Analysis of the sediment trap materials>>> analysis of two sediment trap material

---

## Author Comment (AC1) · 23 Feb 2018

**Supplement. Answer to reviewer #1**

We thank anonymous referee #1 for the interest he showed in our manuscript and detailed comments that have helped to improve the original version of the manuscript. We have considered all his/her comments and addressed each of his/her concerns below.

*R1-Cx : Referee comment*, **R1-Rx:** authors response.

**R1-C1:** *The manuscript documents an annual record of coccolithophore production and coccolith weights/lengths at a Southern Ocean site. The topic is worthy of publication in biogeosciences, and the results will be of potential interest to the wider scientific community.*
*In general, the manuscript is well written and illustrated, and does not contain any major flaws. However, see below for minor points.*
*Scientific points Line 38: " : : : coccolith assemblages experienced weight and length reduction .." !? It is not the assemblages that have reduced weight and length, it is the coccoliths*
**R1-R1:** Corrected according to reviewer 1' suggestion. Now it reads: "coccoliths captured by the traps experienced weight and length…"

**R1-C2:** *Line 89: the Southern Ocean is a small area ? – 25% of the global area looks quite significant to me*
**R1-R2:** The sentence referred by reviewer #1 has been rephrased in order to avoid subjective descriptions of the size of the Southern Ocean. Now it reads: "Despite the fact that the Southern Ocean accounts for about 25% of the global ocean, it contains ~40% of the global ocean inventory of anthropogenic $CO_2$"

**R1-C3:** *Line 148: why is this section in the methods ? move regional setting and oceanography to the introduction*
**R1-R3:** Corrected according to reviewer 1' suggestion. In the new version of the manuscript, section "Regional setting and oceanography" has been moved to the introduction (subsection 1.2). Section 1.1. has been titled "1.1. Background and objectives". Subsections of Material and Methods section have been renumbered accordingly.

**R1-C4:** *Line 235: why unfiltered ? I am not a specialist on sediment traps but it seems odd to use unfiltered seawater. Is there not a risk of contamination ?*
**R1-R4:** The water used to fill the sediment trap cups was unfiltered deep seawater from > 1000m, where the particle abundance is so low that filtering is unnecessary and hard to do without adding more particles than you remove. Moreover, it is important to highlight that the risk of contamination is negligible since the particle levels in sea water are of the order of micrograms per litre while concentration in the trap cups after recovery are of the order of milligrams per litre. This point has been clarified in the new version of the manuscript. The following text has been added in lines 446-449 of the new version of the manuscript with tracked changes:
"Risk of sample contamination by the unfiltered seawater is considered negligible due to the fact that the deep water exhibits low particle abundance and also because particle concentration in sea water is of the order of µg/L while concentration in the trap cups after recovery was of the order of mg/L."

**R1-C5:** *Lines 357 and 345: "coccolith particle bloom" – since coccoliths are inanimate (just pieces of calcium carbonate) I think the word 'bloom' is inappropriate here – use 'The summer coccolith flux exhibited : : :'*

**R1-R5:** Corrected according to reviewer 1's suggestion.

***R1-C6:*** *Line 548: you mention two factors that possibly explain the changes in calcification. Calcification (i.e. overgrowth) tends to increase with depth in the photic zone, at least in some areas of the world. So in winter it may be that the coccolithophores are sitting deeper and therefore have more calcified coccoliths than in the summer when they are closer to the surface and therefore with lightly calcified coccoliths. Of course this difference in surface vs deeper photic could be related to various parameters (light, nutrients, temperature). Do you have data/images of coccoliths from different photic depths ? In Plate 1 you show lightly and heavily calcified coccoliths from the traps – but how do they relate to the surface oceans ?*

**R1-R6:** We appreciate reviewer 1's comment. Unfortunately, there is no data available of coccolith weight from different photic depths. Only samples collected by two sediment traps (that were deployed far below the photic zone) and satellite data are available for the study site. Therefore, our current data set precludes the assessment of the relationship between coccolith weight and the depths were the coccolithophore populations developed. In regard to seasonality, no relationship between the overgrowths and a particular period of the annual cycle was observed. This is now clarified in a sentence that has been included in the new version of the manuscript with tracked changes ("lines 674-675").

**Minor points for correction/consideration**

***R1-C7:*** *Title and elsewhere: Just a query. Is the use of Australian Sector OK ? Naming the sectors after the oceans, like the Atlantic Sector, Pacific Sector and Indian Ocean Sector is fine, but I wonder whether using country names (for sectors and territories) is considered to be geopolitical.*

**R1-R7:**

We acknowledge the point highlighted by reviewer 1. Indian sector could also be an appropriate term for referring to the study region of this research. Nonetheless, we decided to use the term "Australian sector of the Southern Ocean" in order to be consistent with the terminology of previous work along the 140°E parallel such as Findlay and Giraudeau (2000), Quéguiner (2001), Trull et al. (2001), Sedwick et al. (2008), de Salas et al. (2011), Lannuzel et al. (2011) and many others. Please find the references of the publications mentioned in the previous sentence listed below:

de Salas, M. F., Eriksen, R., Davidson, A. T., and Wright, S. W.: Protistan communities in the **Australian sector** of the Sub-Antarctic Zone during SAZ-Sense, Deep Sea Research Part II: Topical Studies in Oceanography, 58, 2135-2149, http://dx.doi.org/10.1016/j.dsr2.2011.05.032, 2011.

Findlay, C. S., and Giraudeau, J.: Extant calcareous nannoplankton in the **Australian Sector** of the Southern Ocean (austral summers 1994 and 1995), Marine Micropaleontology, 40, 417-439, http://dx.doi.org/10.1016/S0377-8398(00)00046-3, 2000.

Lannuzel, D., Bowie, A. R., Remenyi, T., Lam, P., Townsend, A., Ibisanmi, E., Butler, E., Wagener, T., and Schoemann, V.: Distributions of dissolved and particulate iron in the sub-Antarctic and Polar Frontal Southern Ocean (**Australian sector**), Deep Sea Research Part II: Topical Studies in Oceanography, 58, 2094-2112, http://dx.doi.org/10.1016/j.dsr2.2011.05.027, 2011.

Quéguiner, B.: Biogenic silica production in the **Australian sector** of the Subantarctic Zone of the Southern Ocean in late summer 1998, Journal of Geophysical Research: Oceans, 106, 31627-31636, 10.1029/2000JC000249, 2001.

Sedwick, P. N., Bowie, A. R., and Trull, T. W.: Dissolved iron in the **Australian sector** of the Southern Ocean (CLIVAR SR3 section): Meridional and seasonal trends, Deep Sea Research Part I: Oceanographic Research Papers, 55, 911-925, http://dx.doi.org/10.1016/j.dsr.2008.03.011, 2008.

Trull, T. W., Bray, S. G., Manganini, S. J., Honjo, S., and François, R.: Moored sediment trap measurements of carbon export in the Subantarctic and **Polar Frontal zones of the Southern Ocean, south of Australia**, Journal of Geophysical Research: Oceans, 106, 31489-31509, 10.1029/2000JC000308, 2001.

*R1-C8: Line 30: Don't mix z and s verbs. For example, here you use 'characterized' and on line 151 'summarized', but on line 35 you use 'analysed' and on line 135 'fertilisation'. Furthermore, on line 236 you use 'programmed' and on line 349 'grey'. You need to be consistent, and choose between British English and US English. It looks like you are favouring the former.*
**R1-R8:** Corrected according to reviewer 1's suggestion. The whole manuscript has been revised and corrected in order to be consistent with the use of British English (i.e. verbs "z" has been replaced by "s" when needed".

*R1-C9: Line 45: coccolithophorid vs coccolithophore. Be consistent, and choose one.*
**R1-R9:** Corrected according to reviewer 1's suggestion. In order to be consistent the word coccolithophorid has been replaced by coccolithophore in the new version of the manuscript.

*R1-C10: Line 62: " ..some species (but not all) of coccolithophore .." – please change to 'some species of (but not all) coccolithophores ..'*
**R1-R10:** Corrected according to reviewer 1's suggestion.

*R1-C11: Lines 71-75: needs to be rewritten, as it doesn't make sense*
**R1-R11:** The sentence highlighted by reviewer 1 have been deleted following the suggestion of reviewer 2 (See **R2-C21).**

*R1-C12: Line 79: one bracket is missing*
**R1-R12:** Corrected according to reviewer 1's suggestion.

*R1-C13: Line 102: dominantly present -> dominate*
**R1-R13:** Coccolithophores are abundant in the subantartic waters of the Southern Ocean, but this does not mean that they dominate the phytoplankton communities in terms of numbers or biomass. In order to be more precise, the sentence highlighted by reviewer 1 has been rewritten: "coccolithophores exhibit high concentrations in the Subantarctic Southern Ocean"

*R1-C14: Line 115: spares -> sparse*
**R1-R14:** Corrected according to reviewer 1's suggestion.

*R1-C15: Line 149: "..at the north : : : at the south .." -> ' ..in the north : : : in the south ..'*
**R1-R15:** Corrected according to reviewer 1's suggestion.

*R1-C16: Line 203: CO2 rich -> CO2-rich*
**R1-R16:** Corrected according to reviewer 1's suggestion.

*R1-C17: Line 253: "After settling 12 hours .." -> 'After settling for 12 hours ..'*
**R1-R17:** Corrected according to reviewer 1's suggestion.

*R1-C18: Line 271: by using-> using*
**R1-R18:** Corrected according to reviewer 1's suggestion.

*R1-C19: Line 273: " ..to the winter .." -> ' ..to winter ..'*
**R1-R19:** Corrected according to reviewer 1's suggestion.

*R1-C20: Line 285: Scanning Electron Microscopy -> scanning electron microscope (SEM)*

**R1-R20:** Since this point has been also mentioned by reviewer 2 (See **R2-C55**), the text has been modified trying to satisfy both reviewers suggestions. The text now reads: "a Scanning Electron Microscope (SEM)".

*R1-C21: Line 298: "..using a with a Nikon .." -> ' : : :using a Nikon ..'*
**R1-R21:** Corrected according to reviewer 1's suggestion.

*R1-C22: Line 346: " ..should be looked with caution .." -> 'should be viewed with caution'*
**R1-R22:** Corrected according to reviewer 1's suggestion.

*R1-C23: Line 358 (and elsewhere) : you need to insert x (times) between the number and the power. For example, 2.2 10 -> 2.2. x 10*
**R1-R23:** Corrected according to reviewer 1's suggestion. The manuscript has been revised and "x" has been included between the number and the power when absent.

*R1-C24: Line 365: Biogenic -> biogenic*
**R1-R24:** Corrected according to reviewer 1's suggestion.

*R1-C25: Line 370: of the species Calcidiscus -> of Calcidiscus*
**R1-R25:** Corrected according to reviewer 1's suggestion.

*R1-C26: Figure 5 (and elsewhere): I realise that 'liths' is in common use in presentations, but it is not an official term. Better to use coccoliths.*
**R1-R26:** Corrected according to reviewer 1's suggestion. The word "liths" has been replaced by coccoliths in the Y-axis of Figure 5. The text has been revised although no inconsistencies were found.

*R1-C27: Line 382: tiles? I think you mean 'elements'*
**R1-R27:** PREGUNTAR A Lluïsa Cross The text has been corrected following reviewer 1's recommendation. The term tiles has been replaced by "tile-like elements".

*R1-C28: Line 384: "Distal shield measures ranged between 2 and 4,35 .." -> 'Distal shield ranges from 2.0-4.35 ..' [use decimal point not comma]*
**R1-R28:** Corrected according to reviewer 1's suggestion.

*R1-C29: Line 424: here you use station 62 S, and before 62 S site – perhaps be consistent in usage*
**R1-R29:** The words site and station are used as synonyms in the text and are used alternatively in order to avoid repetition. Therefore, no changes in the usage of these words have been incorporated in the text.

*R1-C30: Line 463: genetical -> genetic*
**R1-R30:** Corrected according to reviewer 1's suggestion.

*R1-C31: Line 550: degrees of calcification -> degree of calcification*
**R1-R31:** Corrected according to reviewer 1's suggestion.

*R1-C32: Line 555: B/C south 50_S -> B/C south of 50_S*
**R1-R32:** Corrected according to reviewer 1's suggestion.

*R1-C33: Line 576: light-dependant -> light-dependent*
**R1-R33:** Corrected according to reviewer 1's suggestion.

*R1-C34: Line 617: absence accompanying in situ -> absence of in situ*
**R1-R34:** Corrected according to reviewer 1's suggestion.

***R1-C35:*** *Line 643: That supported -> That is supported*
**R1-R35:** Corrected according to reviewer 1's suggestion.

***R1-C36:*** *Lines 741-759: delete, as the same as later references*
**R1-R36:** Corrected according to reviewer 1's suggestion.

***R1-C37:*** *Lines 773 and 816 (and elsewhere): Deep Sea -> Deep-Sea*
**R1-R37:** Although we agree with reviewer 1 that the title of the journal is "Deep-Sea Research" all the references downloaded from their official website display the title of their own journal like "Deep Sea Research". Therefore, in order to be consistent with the references of the journal we have kept the title in all references without the "-".

***R1-C38:*** *Line 860: emiliania huxleyi*
*(haptophyta) ´z, Journal of phycology, -> Emiliania huxleyi 860 (haptophyta) ´z,*
*Journal of Phycology, [why is there a superscript 1 at the end of the title ?*
**R1-R38:** Corrected according to reviewer 1's suggestion. The superscript 1 has been deleted from the title.

***R1-C39:*** *Lines 864, 867 and 884 ( and elsewhere): italicize the species name*
**R1-R39:** Corrected according to reviewer 1's suggestion. The references have been revised and species names have italicized.

**Additional changes**

- Section 3.3, line 674, of the "new version of the manuscript with tracked changes", the sentence "annual amplitude of the coccolith weight was approximately" has been replaced by "annual amplitude of the **mean** coccolith weight was approximately" in order to be clearer.
- The correlation between Biogenic silica and coccolith fluxes at 2000 m showed in line XX of the first version of the manuscript was the correlation coefficient (r = 0.86), not the coefficient of determination ($R^2$ = 0.74). In the new version of the manuscript the coefficient of determination is shown.
- The coccolith length values presented the results listed in section 3.3 of the first version of the manuscript (lines 405-409) corresponded to an earlier version of the data set.. The correct values have been included in the new version of the paper. Please note that the seasonal trend remains identical (only there was is a slight variation of the absolute values). Please also note that the calculations made in the discussion regarding the relationship between size and weight in the first paragraph of Section 4.3 are correct. The coccolith length data at 3700 plotted in figure 6 also corresponded to the older version of the dataset. This has been corrected in the new version of the manuscript. Please note that the seasonal trend remains identical.

- Plate I: In the first version of the manuscript the we skipped the letter "d" listing the photos of Plate I. In the new version of the manuscript, this typo has been fixed.

---

## Author Comment (AC2) · 23 Feb 2018

**Supplement. Answer to reviewer #2**
*R2-Cx : Referee comment*, **R2-Rx:** authors response.

*R2-C1: **General comments.***
*The calcification of coccolithophores in the high latitude regions is a growing concern as it will have large influence on ocean biogeochemistry and thus climate. The data provides new information on coccolithophore response to varying environmental conditions at Antarctic Zone using sediment traps samples. However, the information provided here can be expressed in a much better way. Sediment trap data from Southern Ocean is difficult to obtain and is therefore a welcome addition to knowledge and needs to be published. Here are some major and minor comments which need to be incorporated in the manuscript.*

**R2-R1:** We sincerely thank reviewer #2 for the careful reading of our manuscript and constructive criticisms and comments that helped to improve the manuscript. The text and figures have been revised and improved accordingly. Next, we briefly summarize the main changes included in the text. A potential explanation behind the formation of secondary crystallizations observed in some of the coccoliths has been included together with the references suggested by the reviewer. Also, in the new version of the manuscript the possible influence of salinity on coccolith morphology is discussed and new references dealing with the impact of temperature in the coccoliths have been included. Moreover, several figures have been improved following reviewer 2's suggestion: Figure 1 now shows the maximum and minimum sea ice extent during the deployment period; the vertical structure of temperature of the water column in Figure 2 is now plotted with Ocean Data View; several aspects of Figure 3 have been improved following reviewer 2's comments; and a new figure has been created (Fig. 7) that shows the regression plots between coccolith weight and length. Finally, a throughout revision of the references cited in the manuscript has been performed.

*R2-C2: **Major comments/changes needed to be done:***
*1. Authors documented only abundance of coccoliths of E. huxleyi B/C morphotypes in the traps. It is also mentioned that occurrence of other coccolithophores were also documented. Though other coccoliths are in low abundance, it represents changes in the upper oceanic conditions. It is also important to plot a graph of other coccolithophores and discuss what their assemblage indicates.*
**R2-R2:** Due to the very low abundance of other coccolithophore species in the trap samples, authors decided to focus the discussion on total coccolith fluxes because flux plots based on very small counts could be biased and therefore misleading for the reader. However, in compliance with reviewer's request the fluxes and relative contribution of *Emiliania huxleyi*, *Calcidiscus leptoporus* and *Gephyrocapsa* spp. at both sediment trap depths have been included in the new Supplementary Figure 1. Moreover, the seasonality of these species is now described in the section 3.1 of the results, as well as, discussed at the end of the first paragraph of section 4.2.

*R2-C3: 2. The overcalcification of E. huxleyi is documented by few researchers in past few years. But not many papers are published on this. Authors have documented overcalcification on coccoliths retrieved from sediment traps. I assume intact coccospheres are also documented in the both the sediment traps. In this case, whether authors documented overcalcification on coccospheres of E. huxleyi? If any information*

*is available on living coccolithophores in this region, it should be included. It is important to document the overcalcification on E. huxleyi is a natural process and not a part of secondary calcification. So, if extant coccolithophores data is available at/around study site then it should be presented.*

**R2-R3:** We appreciate reviewer 2's comment on the possible overcalcification of some of the coccoliths captured by the trap. As stated in section "3.2 SEM analyses", the unusual structures (mainly small spherules deposited on the coccolihts) observed on some of the coccoliths, such as that of the coccoliths shown in Plate I, e-g, is attributed to a secondary recrystallization but not overcalcification. This interpretation is based on the small spherules often present in the coccoliths, particularly on the laths, a feature consistent with a secondary recrystallization and not with overcalcification of the coccoliths during the life cycle of the coccolithophore.

Cubillos et al. (2008) undertook a comprehensive analysis of the *E. huxleyi* morphotypes in the Australian sector of the Southern Ocean along the 140°E meridian (covering the location of the sediment trap station analysed here). According to the former authors the overcalcified forms of *Emiliania huxleyi* are restricted north of the Polar Front (Subantarctic and Subtropical Zones), and therefore it is unlikely that we register these forms in our traps. Furthermore, the only overcalcified forms reported by Cubillos et al. (2008) correspond to morphotype A, which is characterized by a larger coccoliths than those observed here. Please note that Cubillos et al. (2008) paper is discussed in section 4.3 of the manuscript. Moreover, an explanation about the possible origin of the "small spherules" observed on some coccoliths has been included at the end of section 4.4. of the discussion in order to satisfy reviewer 2's request.

In regard to the documentation of coccospheres in the trap samples. The number of coccospheres found in the samples was very low due to the low abundance of coccolithophores in the study region. In order to overcome this problem, one could state that concentration used in the SEM preparations could have been increased. However, this was not possible in our particular case due to the large abundance of diatoms at this site which concentration obviously increase with concentration. Please note that the biogenic silica fluxes at the 61ºS site are the arguably the highest ever reported in the world's ocean. Such high concentration of diatoms greatly hampered the finding of coccospheres under the SEM, that it is the only methodology that allows discriminating between overcalcified and "normal" coccospheres.

*R2-C4: 3. Authors often compared diatom assemblage with coccoliths. Plotting a graph of total diatom assemblage vs coccoliths abundance will be useful for understanding changes in the AZ region.*

**R2-R4:** Authors do not completely understand reviewer 2's request. Total diatom valve and total coccolith fluxes at both sediment trap depths are plotted in figure 5. Only one diatom species is mentioned in the text: *Thalassiotrix antarctica*. The seasonal succession of diatom species at the 61ºS site is discussed in detail in Rigual-Hernández et al. (2015, JMS) paper that is mentioned in the text. Authors believe that plotting the fluxes and relative abundance of the all the diatom species would be out of the scope of this paper, would not contribute to the discussion and would be misleading for the reader. However, these graphs can be included in the manuscript or in a supplementary figure if the editor considers this information relevant.

*R2-C5: 4. A recent study indicates polysaccharides are also responsible for overcalcification of coccolithophres. Authors need to discuss outcomes in more detail and should be cited with recent references.*

**R2-R5:** Corrected according to reviewer 2's suggestion. A paragraph dedicated to the possible role of the *polysaccharides serving as* organic scaffold for coccolith formation has been included and new references included and discussed (Gal et al., 2016; Lee et al., 2016) (end of section 4.4)

*R2-C6: 5. Authors should be consistent in framing sentences. Some sentences are too large, some are too small. Be consistent in using AZ vs AZ-S, Fe vs. iron, E. huxleyi, vs Emiliania huxleyi etc.*

**R2-R6:** Corrected according to reviewer 2's suggestions. The whole manuscript has been revised and several sentences have been split into two when possible:

➢ Line 85 of the new version of the manuscript with tracked changes: a long sentence has been removed.
➢ Line 86-87: the sentence has been split into two.
➢ Line 145-146: the sentence has been split into two.
➢ Line 1096-1098: The introductory sentence of the conclusions section has been split into two and rephrased.

Moreover, the text has been revised for inconsistencies in the use of the terms:

➢ AZ vs AZ-S, Fe vs. iron, *E. huxleyi* vs *Emiliania huxleyi*.
➢ Fe vs. iron: The term "Fe" has been replaced by "iron" is the new version of the manuscript.
➢ The term "AZ" has been replaced by "AZ-S" when possible.
➢ *Emiliania huxleyi* has been replaced by *E. huxleyi* when possible.

*R2-C7: 6. Since, both sediment traps are located in the 61degS and there is no comparison done with other sediment traps showed in the Fig. 1, it is not necessary to mention "at 61S sediment trap everywhere"*

**R2-R7:** Corrected according reviewer 2's suggestion. The name of the sampling site has been replaced by synonyms when possible. However, the term 61°S site is still used in the new version of the manuscript when needed.

*R2-C8: 7. Authors should crosscheck references very carefully. Many references listed in the text are not reflected in the reference list. Similarly, many references listed in the Reference list are not*

**R2-R8:** Corrected according to reviewer 2's suggestion. The references listed in the text and in the reference list have been revised. Only a few errors were found and have been corrected in the new version of the manuscript.

*R2-C9: mentioned in the text. Genus and species name should appear properly and in italics.*

*For ex. Line 860- emiliania huxleyi.*
**R2-R9:** Corrected according to reviewer 2's suggestion. The text and references have been thoroughly revised in order to show all species and genus names in italics.

*R2-C10:Hagino et al. 2011 reference- written in caps*
**R2-R10:** Corrected according to reviewer 2' suggestion.

*R2-C11:**Minor comments:***
*Line 14-30: The information provided in the abstract can be shifted to the introduction. The abstract should start from Line 31.*
**R2-R11:** We agree with reviewer 2 that the first lines of the abstract could also fit in the introduction. However, we believe that providing a very short rationale of the experiment and highlighting the gaps in the knowledge about the effects of a changing climate in Southern Ocean ecosystems is important to help the non-specialized reader to understand the relevance of our study, thereby potentially reaching a larger audience. Therefore, authors have decided to leave the abstract as it is in the first version of the manuscript.

*R2-C12:Line 32: In the deep ocean >>> at the Antarctic Zone*
**R2-R12:** Corrected according to reviewer 2's suggestion. Now it reads: "We report here on seasonal variations in the abundance and composition of coccolithophore assemblages collected by two moored sediment traps deployed at the Antarctic Zone south of Australia (2000 and 3700 m depth) for one year in 2001-02."

*R2-C13:Line 33: ~2000>>> 2000*
**R2-R13:** Corrected according to reviewer 2's suggestion.

*R2-C14:Line 33: and 3700>>> and 3700 respectively*
**R2-R14:** In the new version of the manuscript this sentence has been rephrased. As consequence of this change is no need to include the word "respectively".

*R2-C15:Line 37: Emiliania huxleyi morphotype B/C>>>> E. huxleyi morphotype B/C*
**R2-R15:** Corrected according to reviewer 2's suggestion.

*R2-C16: Line 38: coccolith assemblage experienced weight and length reduction>>>*
*coccoliths experienced weight and length reduction*
**R2-R16:** This sentence had already been corrected following reviewer 1's suggestion.

*R2-C17: Line 39: during the summer months>>> during summer months (December-March?)*
**R2-R17:** The text has been modified slightly different to that suggested by reviewer 2. Now it reads: "reduction during summer (December – February)"

*R2-C18: Line 40: at both sediment trap depths>>> at both sediment traps*
**R2-R18:** Corrected according to reviewer 2's suggestion.

*R2-C19: Line 41: in other southern ocean settings>>>> which settings?*
**R2-R19:** Corrected according to reviewer 2's suggestion. Patagonian shelf and Scotia sea are mentioned between brackets in the new version of the manuscript.

*R2-C20: Line 43-46: Apart from first record, significant outcomes of the study needs to be highlighted here.*
**R2-R20:** The major findings of the investigation are summarized (and numbered) before this sentence. Moreover, in the last sentence of the abstract we clearly explained the importance of our results clearly explaining that our results provide a reference/baseline for evaluation of Southern Ocean coccolithophore responses to changing environmental conditions in the coming decades. Therefore, we believe that the main objectives and outcomes of the study are already mentioned in the text and no more extra information is required.

*R2-C21: Line 71-75: May not required in the introduction*
**R2-R21:** Corrected according to reviewer 2's suggestion. The text: "For example, diatoms can play a prominent role in export of organic matter from the surface ocean, because of their heavy siliceous frustules and capacity for aggregation and rapid sinking facilitates efficient transport of organic carbon (Buesseler, 1998; Smetacek, 1999). Nonetheless, it has also been suggested that this silica-mediated carbon export driven by diatoms may not always reach the ocean interior efficiently (Francois et al., 2002; Lam and Bishop, 2007)." has been removed from the introduction.

*R2-C22: Line 76: Coccolithophores also has the potential >>>> coccolithophores has potential*
**R2-R22:** Corrected according to reviewer 2's suggestion.

*R2-C23: Line 89: 25% of ocean area is not small.*
**R2-R23:** This sentence has been modified following the suggestion of both reviewers. Now it reads: "Despite the fact that the Southern Ocean accounts for about 25% of the global ocean, it contains ~40% of the global ocean inventory of anthropogenic $CO_2$"

*R2-C24: Line 107: Trull et al., 2017 is not mentioned in reference list*
**R2-R24:** We believe there must have been a misunderstanding here since Trull et al., 2017 (Biogeosciences) was mentioned in the first version of the manuscript (lines 1097-1099)

*R2-C25: Line 109: Cubilos et al. 2008 or 2007? Cubilos et al. 2008 is not mentioned in the reference list*
**R2-R25:** Corrected according reviewer 2's suggestion. Cubillos et al. 2008 does not exist. The text has been corrected, now only the citation Cubillos et al., (2007) appears in the text.

*R2-C26: Line 111: Freeman and Lovenduski (2015) not mentioned in the reference list*
**R2-R26:** Corrected according reviewer 2's suggestion. The reference Freeman and Lovenduski (2015) appears now in the text.

*R2-C27: Line 116-122 and elsewhere in the introduction: very large sentence. Authors should be consistent in framing sentences. Such long sentences to be avoided in the introduction*
**R2-R27:** Corrected according reviewer 2's suggestion. Please see **R2-R6** for more details.

*R2-C28: Line 123: Trull et al., 2017 missing in the reference list*

**R2-R28:** As mentioned in a previous comment (**R2-R24**), we believe that there must have been a misunderstanding here since Trull et al., 2017 did appear in the reference list (lines 1097-1099 of the first version of the manuscript)

*R2-C29: Line 138: inferred from one-year record>>>> inferred from 10 month record*
**R2-R29:** The sentence has been rephrased and the words "one-year record" replaced by "during ten months".

*R2-C30: Line 139: SOIREE……. Elaborate when using short forms for the first time*
**R2-R30:** Corrected according to reviewer 2's suggestions.

*R2-C31: Line 147: Regional setting and oceanography; and water carbonate chemistry, should be shifted in the introduction under a different sub-heading.*
**R2-R31:** Corrected according to both reviewer 2 and reviewer1's suggestion. Sections "regional setting and oceanography" is section 1.2 in the new version of the manuscript, while "Water column chemistry in the study region" is now section 1.3

*R2-C32: Line 151… SAACF>>>SACCF*
**R2-R32:** Corrected according to reviewer 2's suggestion.

*R2-C33: Line151: (SAACF)>>>(SACCF) Fig.1*
**R2-R33:** Corrected according to reviewer 2's suggestion. Figure 1 in now cited between brackets at the end of the first sentence of section "1.2. Regional setting and oceanography ".

*R2-C34: Line 155: upper water column with nutrients (add reference). Similarly for sentences between lines 155-160 (add references)*
**R2-34:** Before the description of the seasonal evolution of water column physical, chemical and biological properties, it is mentioned: "Trull et al. (2001b) summarized the seasonal evolution of water column properties in the study region". Although we could refer to this cite in each following sentence, authors believe that the text should remain as is now in order to avoid repetition.

*R2-C35: Line 155: Chlorophyll-a, vs/ Line 484 chl-a>>> use one style of writing*
**R2-R35:** Corrected according reviewer 2's suggestion. The term chl-*a* has been replaced by chlorophyll-*a* and the text has been revised for inconsistencies.

*R2-C36: Line 160: ….. in algal biomass occurs within the mixed layer (add reference)*
**R2-R36:** Corrected according reviewer 2's suggestion. The reference Trull et al. (2001b) has been included.

*R2-C37: Line 164: Trull et al. 2001>>>> Trull et al. 2001a or Trull et al. 2001b or 2001c??*
**R2-R37:** Trull et al. 2001b is the correct reference here. In the new version of the manuscript the correct reference is specified.

*R2-C38: Line 169: Large calanoid copepodites.>>> Copepods and copepodites are different. Copepodites are immature form of copepods. What authors are trying to say? "large Calanoid copepods" or "mainly Calanoid copepodites"??*

**R2-R38:** As highlighted by reviewer 2, copepodites are inmature forms of copepods. Zeldis et al. (2001) reported that "The SOIREE site mixed-layer mesozooplankton community was dominated by copepods, with salps and pteropods absent, and euphausiids either absent or very rare (maximum 7 animals m$^{-3}$). The copepod community was numerically dominated by large copepodites (> 1.5mm prosome)…". Therefore, we believe that the sentence "Mesozooplankton analysis during the SOIREE experiment by Zeldis (2001) indicates that zooplankton community in the study region is dominated by copepods, mainly large calanoid copepodites." is correct. That is the reason why the sentence has not been modified.

*R2-C39: Fig. 1. Since author has mentioned that sediment trap location was away from sea ice activity, authors should draw seasonal sea ice zone or winter sea ice limit for the year 2001-2002 in Fig. 1*
**R2-R39:** Corrected according to reviewer 2's suggestion. The winter sea ice limit for August 2001 is now showed in Figure 1. The Figure caption has been adapted accordingly citing the source of the sea ice data represented in Fig. 1. Moreover, the database where this data was obtained is also cited in the new version of the manuscript.

*R2-C40: Line 180: SACCF- Southern ACC Front>>>> SACCF- Southern Antarctic Circumpolar Current Front*
**R2-R40:** Corrected according to reviewer 2's suggestion.

*R2-C41: Line 193: calcite saturation horizon (CSH)>>>> Calcite Saturation Horizon (CSH)*
**R2-R41:** Corrected according to reviewer 2's suggestion.

*R2-C42: Line 194: CaCO3 compensation depth (CCD)>>>> CaCO3 Compensation Depth (CCD)*
**R2-R42:** Corrected according to reviewer 2's suggestion.

*R2-C43: Fig. 2. Similar to Fig. 1 and Fig. 3, Fig. 2 should be plotted in the Ocean data view and figures should be readable. What does the dotted line indicate in Fig. 2b?*
**R2-R43:** Corrected according to reviewer 2's suggestion. The vertical structure of temperature of the water column has been plotted in Ocean Data view (Fig 2a of the new version of the manuscript). Due to the low number of observations of nutrient concentrations in the study regions, their representation using ODV would require a large interpolation of measurements and the resulting graph would be somewhat misleading for the reader. Therefore, authors have decided to leave the Figure representing silicate and nitrate concentration as it was in the first version of the manuscript (Fig. 2b in the new version of the manuscript).

*R2-C44: Fig 2 a. legend should contain surface macronutrient concentrations?*
**R2-R44:** Only data of the nutrient concentration in the mixed layer is available. This data is data plotted in Figure 2b. In order to be clearer, in the new version of the manuscript it is clarified that the data showed in Figure 2b is representative for the mixed layer.

*R2-C45: Line 200: (Fig. 3)>>> (Fig.3a) or (Fig. 3b)*
**R2-R45:** Corrected according to reviewer 2's suggestions. Figure 3b is mentioned in the new version of the manuscript.

*R2-C46: Fig. 3. Mark Fig. 3a and Fig 3b; Mark frontal locations, put units for color scale bar, x axis etc. Also, include sampling dots if possible. Mark 1000m sediment trap in fig 3 in different color., as it is mentioned in line 226.*
**R2-R46:** Figure 3 has been corrected according to reviewer 2's suggestions.

*R2-C47: Line 210: Elaborate when appear for the first time*
**R2-R47:** Corrected according to reviewer 2's suggestions.

*R2-C48: Line 211: Tanhua et al. 2008 is missing in the reference list*
**R2-R48:** The reference Tanhua et al. (2008) has been replaced by CARINA group (2011) which appears now in the reference list. The new reference refers to the same data set of that used by Tanhua et al. (2008), i.e. both references are correct.

*R2-C49: Line 122: Draw seasonal sea ice zone in Fig. 1*
**R2-R49:** Corrected according to reviewer 2's suggestion. In the new version of the manuscript the Maximum Winter Sea Ice Extent and Minimum Summer Ice Extent for the study period (August 2001 and February 2002) are represented in Figure 1. The Figure caption has been rewritten accordingly and the dataset from where the sea ice data has been extracted is now cited in the text (Fetterer et al., 2017).

*R2-C50: Line 225: for approximately one year>>>> for ten months*
**R2-R50:** Strictly speaking the sampled period is 10 months and a half (317 days / 30 day per month = 10.56 months). Authors believe that it is correct to leave the text as it is now because it is specified the number of days sampled between brackets. "The 61°S mooring was equipped with three McLane Parflux time series sediment traps (Honjo and Doherty, 1988) for approximately one year (November 30, 2001 to September 29, 2002, 317 days)."

*R2-C51: Line 227: highlight Fig. 3a and 3b*
**R2-R52:** Corrected according to reviewer 2's suggestion. In the new version of the manuscript both Fig. 3a and 3b are mentioned in the sentence referred by reviewer 2.

*R2-C52: Line 231: be consistent in using depths. Either use "~2000" or "2000",,,, "3800 or 3700*
**R2-R52:** Corrected according reviewer 2's suggestion. In the new version of the introduction only "2000 m" is used (i.e. not "~2000" used once in the introduction in the first version of the manuscript. Moreover 3800 has been replaced by 3700 following the comments of the reviewer.

*R2-C53: Line 283: Emiliania huxleyi>>>> E. huxleyi*
**R2-R53:** Corrected according to reviewer 2's suggestions.

*R2-C54: Line 235: why unfiltered seawater used? Won't it contaminate samples?*
**R2-R51:** A similar question was raised by reviewer 1 (*R1-C4*). In the new version of the manuscript it has been clarified the reasons why unfiltered seawater was used. In the new version of the manuscript it is stated: "Risk of sample contamination by the unfiltered seawater is considered negligible due to the fact that the deep water is characterized by low particle abundance and also because particle concentration in sea water is of the order of µg/L while concentration in the trap cups after recovery was of the order of mg/L."

*R2-C55: Line 285: Scanning Electron Microscopy>>>> Scanning Electron Microscope (SEM)*
**R2-R55:** Corrected according to reviewer 2's suggestion.

*R2-C56: Line 286: decantation method outlined by>>>> method outlined by*
**R2-R56:** Corrected according to reviewer 2's suggestion.

*R2-C57: Line 287: coated in Gold>>> coated with/using Gold*
**R2-R57:** Corrected according to reviewer 2's suggestion.

*R2-C58: Line 288: please mentioned magnification range (for example 2000-7000x) used during analysis.*
**R2-R58:** Corrected according to reviewer 2's suggestion. The magnification used during the SEM analysis is specified in the new version of the manuscript (magnification 5000-20000x).

*R2-C59: Line 288: Emiliania huxleyi>>>>E. huxleyi*
**R2-R59:** Corrected according to reviewer 2's suggestion.

*R2-C60: Line 291: Emiliania huxleyi>>>>E. huxleyi*
**R2-R60:** Corrected according to reviewer 2's suggestion.

*R2-C61: Line 293: Emiliania huxleyi>>>>E. huxleyi*
**R2-R61:** Corrected according to reviewer 2's suggestion.

*R2-C62: 293-294: Emiliania huxleyi coccoliths into morphotypes>>> Emiliania huxleyi into different morphotypes*
**R2-R62:** Corrected according to reviewer 2's suggestion.

*R2-C63: Line 298: using a with a Nikon>>>> using a Nikon*
**R2-R63:** Corrected according to reviewer 1 and 2's suggestions.

*R2-C64: Line 319: sea surface temperature (SST) already elaborated in line 153*
**R2-R64:** Corrected according to reviewer 1 and 2's suggestions.

*R2-C65: Line 320: Sea Surface Temperature Analysis>>>> SST Analysis*
**R2-R65:** Corrected according to reviewer 1 and 2's suggestions.

*R2-C66: Line 323: SST variations>>>> Sea Surface variations*
**R2-R66:** This change has not been incorporated because the authors wanted to refer specifically to Sea Surface Temperatures (SST) not Sea Surface variations in general.

*R2-C67: Line 326: Photosynthetically active radiation (PAR)>>>> Photosynthetically Active Radiation (PAR)*
**R2-R67:** Corrected according to reviewer 2's suggestion.

*R2-C68: Line 327: particulate inorganic carbon (PIC)>>>> Particulate Inorganic Carbon (PIC)*

**R2-R68:** Corrected according to reviewer 2's suggestion. Moreover, in the caption of figure 4, the names Photosynthetically Active Radiation and Particulate Inorganic Carbon have been replaced by their acronyms, i.e. PAR and PIC.

*R2-C69: Fig. 4. Authors have mentioned March as a late summer months in the line 158. In this case, the shaded area should also include March*
**R2-R69:** The sentence highlighted by reviewer 2 has been corrected. In the new version of the manuscript, it reads: "By late summer-early autumn (March) SST ranges between 2 and 3 °C"

*R2-C70: Line 341: particulate inorganic carbon (PIC)>>>> PIC*
**R2-R70:** Corrected according to reviewer 2's suggestion.

*R2-C71: Line 342-346- It is important.……………………... Trull et al., 2017)>>> already mentioned before*
**R2-R71:** Reviewer 2 is right, the fact that the satellite algorithm used to detect PIC is not reliable in Antarctic waters is mentioned in the introduction as well. Nonetheless, authors believe it is important to mention this point again in the caption of Figure 4 in order reinforce this idea and to make the reader aware that the PIC satellite data presented in that figure should be viewed with caution.

*R2-C72: Line 347: particulate organic carbon (POC)>>>> POC; Calcium Carbonate (CaCO3)>>>> CaCO3*
**R2-R72:** Corrected according to reviewer 2's suggestion.

*R2-C73: Fig. 5. Please check scale bars. 2x109 appeared twice on left side.; in the first figure only 109 appeared. Is it 1x109?. Put same scale in both figures. Mark Figure 5a and 5b.*
**R2-R73:** Corrected according to reviewer 2's suggestion. A decimal has been included in the all the labels of axis of Figure 5 in order to avoid the number repetition highlighted by reviewer 2. Moreover, now axis in both figures have the same scale. Finally, the axis titles have been corrected as they were wrongly named in the first version of the manuscript.

*R2-C74: Fig. 5. What is the reason diatom valve flux remained almost constant at 2000 and 3700 but there is an increase in diatom flux during February and March. Please explain.*
**R2-R74:** Differences in the magnitude of fluxes between the upper and deeper trap are most likely due to small differences in the source area of the particles collected by each trap, the so-called statistical funnel (discussed in section 4.1). This statistical funnel increases with depth and therefore it is expected some variability between the fluxes captured by each trap. This is clarified in the new version of the manuscript where it is stated: "The slightly different seasonal pattern observed at both sampling depths (Fig. 5) is mainly attributed to the fact that the area of the ocean from which the particles have been produced increases with depth (Siegel and Deuser, 1997) (863-865 of the new version of the manuscript with tracked changes).

*R2-C75: Line 370-372: which Calcidiscus leptoporus species? Small or intermediate? Which time of sampling month these species are documented?*

**R2-R75**: *Calcidiscus leptoporus* coccoliths were not divided into size classes during the LM microscopy analysis. Therefore, in the new version of the manuscript it is specified "*sensu lato*" after the species name. Relative abundances of this species can be found in Table 1. Moreover, in the new version of the manuscript the fluxes and relative abundance of this species have been plotted in supplementary Figure 1, are described in the results section "3.1 Seasonal dynamics of coccolith export fluxes" and discussed in discussion section "4.2 Seasonal dynamics of the calcareous and siliceous phytoplankton fluxes". Please also note that due to the low abundance of this species in the samples and to its similar seasonal pattern to that of *E. huxleyi*, *C. leptoporus* data does not provide any relevant contribution to the discussion. That is the main reason why initially this data was not included in the first version of the manuscript and also why the information of this species is included as supplement.

*R2-C76: Line 383: distal shields partially missing, may be due to >>>> distal shield partially missing, due to*
**R2-R76**: Corrected according to reviewer 2's suggestion. The sentence has been rephrased to: "…partially missing, mainly due to the slender and delicate structure of the laths".

*R2-C77: Line 384: 2 to 4,35>>>> 2 to 4.35*
**R2-R77**: Corrected according to reviewer 1 and 2' suggestion.

*R2-C78: Line 394: (2.3 and 2.1 pg/coccolith)>>>> (2.3 ±.... and 2.1±.... pg/coccolith)*
**R2-R78**: Corrected according to reviewer 2's suggestion. The standard deviation is now showed in all the coccolith mass values provided in section "3.3 Coccolith weight and length changes"

*R2-C79: Lien 396: (down to 1.6 and 1.9 pg at 2000 and 3700m, respectively)>>>> (down to 1.6±.... and 1.9±.... pg/coccolith at 2000 and 3700m)*
**R2-R79**: Corrected according to reviewer 2's suggestion. The standard deviation is now showed in all the coccolith mass values provided in section "3.3 Coccolith weight and length changes"

*R2-C80: Line 399: Average annual coccolith weight at the 61S traps>>>>it is already mentioned that both traps are located at AZ-S, at 61S. just mentioned depths. Similarly at Line 433, at the 61S site>> study site. Similarly correct changes at line 446 and elsewhere*
**R2-R80**: Corrected according to reviewer 2's suggestion. The name of the station 61ºS site has been replaced when possible in the text. Please not that some time specifying the station is needed, such as in line 446 of the first version of the manuscript. Here we make a comparison with data from other station 47ºS site in the subantarctic zone, and therefore, specifying the name of out sampling site is needed. Please see also **R2-R7**.

*R2-C81: Line 405-406: what makes E. huxleyi coccolith change their lengths in early spring to summer discuss under discussion. Please refer Bollmann et al paper.*
**R2-R81**: Authors believe that reviewer 2 refers to Bollman and Herrle (2007, EPSL) paper where a close relationship between the length of *E. huxleyi* coccoliths and salinity is described. We appreciate reviewer 2's comment and in the new version of the manuscript the possible effect of Sea Surface Salinity (SSS) on the observed coccolith weight and length variability is discussed (lines 966-971 of the corrected version of the manuscript with tracked changes). Moreover, SSS data for the 61ºS site was obtained

from the World Ocean Atlas and is presented in section "2.5 Satellite imagery, meteorological and oceanographic data" of the new version of the publication.

*R2-C82: Line 410-412: If possible, plot graphs of correlation*
**R2-R82**: Corrected according to reviewer 2's suggestion. A new Figure (Fig. 7) has been included in the new version of the manuscript showing the regression plots between *E. huxleyi* coccolith weight and length.

*R2-C83: Line 422: South of the Polar Front>>>> South of the PF*
**R2-R83**: Corrected according to reviewer 2's suggestion.

*R2-C84: Line 422: include recent studies carried out in the Southern Ocean such as, Patil et al., 2017, Saavedra Pellitero et al, Malinverno et al.,*
**R2-R84**: Corrected according to reviewer 2's suggestion. The references suggested by reviewer 2 have been incorporated in the new version of the manuscript.

*R2-C85: Line 426: Buesseler et al., 2007 reference is not in the reference list*
**R2-R85**: Corrected according to reviewer 2's suggestion.

*R2-C86: Line 435: What author mean by "coccolith particle bloom"?*
**R2-R86**: The term "coccolith particle bloom" has been replaced by "the period of enhanced coccolith flux" in order to be clearer.

*R2-C87: Line 450: use either AZ or AZ-S*
**R2-R87**: Corrected according to reviewer 2's suggestion. The term AZ has been replaced by AZ-S in the new version manuscript when required.

*R2-C88: Line 463: general variability they found between>>>> general variability found between*
**R2-R88**: Corrected according to reviewer 2's suggestion.

*R2-C89: Line 463: please differentiate morphological differences between Morphotypes A, B/C, var. huxleyi and var. aurorae. E. hxuelyi morphotype C is usually less calcified morphotype of E. huxleyi usually found in the AZ. What are the probable reasons for absence of morphotype C in sediment trap samples?*
**R2-R89**: Corrected according to reviewer 2's suggestion. A description of the other morphotype found in our study region (morphoptype A) has been included in section 3.2 of the new version of the manuscript with tracked changes (lines 636-656). As stated in the first version of the manuscript (Lines 463-4365), morphotype A has been documented to be genetically different than morphotype B/C. Based on this observation, Cook et al. (2011) associated these two morphologies with two varieties defined as *E. huxleyi* var. *huxleyi* and *E. huxleyi* var. *aurorae*, respectively. The text has been rephrased in order to be clearer (see line 804 of the new version of the manuscript with tracked changes). Findlay and Giraudeau (2000, Mar Mic) and Cubillos et al. (2011) analysed samples from a transect along the 140°E meridian, where our 61°S sediment trap was deployed. Findlay and Giraudeau (2000) did report *E. huxleyi* morphotype C in their samples, Young et al. (2003) reviewed their classification of *E. huxleyi* morphotypes and revised Type C specimens from the Antarctic Ocean documented by Findlay and Giraudeau (2000) to be Type B/C. This is mentioned in Young et al. (2003), Hagino et al. (2005, Mar Mic) and

Cubillos et al. (2007). So the fact that morphotype C has not been previously reported in our study region further supports our observations.

*R2-C90:* Plate 1: I don't agree with all six images belonging to morphotype B/C. Plate 1b, e, looks like morphotype C. Plate 1. g is unrecognizable due to overcalcification. Authors can follow Young et al., 2003 atlas for differentiating E. huxleyi morphotypes.
R2-R90: Authors did follow Young et al. (2003) classification, in fact, one of the co-authors (Lluïsa Cros) co-authored of the Atlas referred to by reviewer 2. Authors believe that the different morphologies observed in the coccoliths are just variations within an E. huxleyi B/C population. Morphotype B/C exhibits a similar morphology to types B and C (Young et al. 2003) but it is intermediate in size. However, coccolith size was considered of limited value in discriminating morphotypes by Cubillos et al. (2007) based on the large variability in size of the coccoliths on the same coccospheres. This is now clearly explained in the paper (Lines 646-656). It is also worth noting that Young et al. (2003) revised the morphotype described as C in our study area by Findlay and Giraudeau (2000) and redefined it as Type B/C.

*R2-C91: Line 508: Silicate and/or Fe>>>> Silicate and/or iron*
**R2-R91**: Corrected according to reviewer 2's suggestion.

*R2-C92: Line 522, line 535: Tagliabue et al., 2014. Year of publication missing in the reference list*
**R2-R92**: We believe there has been a misunderstanding because the year of the publication (2014) is mentioned was in line 1074 of the first version of the manuscript.

*R2-C93: Line 533: the part of the Emiliania huxleyi>>>> the part of the E. huxleyi*
**R2-R93**: Corrected according to reviewer 2's suggestion.

*R2-C94: Line 598: Trull et al., 2001>>>>> Trull et al., 2001a or 2001b or 2001c?*
**R2-R94**: Corrected according to reviewer 2's suggestion. The reference of Trull et al. 2001b is not specified in the text.

*R2-C95: Line 623: both traps (2.5 pg at 2000m to 2.6 pg at 3700)>>>>> both traps (2.5±.... pg at 2000m to 2.6±.... at 3700m)*

**R2-R95**: Since the annual coccolith weights are already mentioned in the results section and later on in the discussion (section 4.5 Calcium carbonate content of *Emiliania huxleyi* coccoliths), the annual coccoliths weights have been removed here in order to avoid repetition.

*R2-C96: Line 634: (Turner, 2002: Turner, 2015)>>>> (Turner, 2002; 2015)*
**R2-R96**: Corrected according to reviewer 2's suggestion.

*R2-C97: Line 655: E. huxleyi morphotype B/C is more weakly calcified than other morphotypes>>>> I am not convinced with this. E. hulxleyi morphotype C is more weakly calcified than B/C. It can be written as >> E. huxleyi morhotype B/C is weakly calcified than A> if authors want to tell extent of calcification.*
        **R2-R97**: In order to be clearer, the sentence highlighted by reviewer 2 has been rephrased. Now it reads" Since *E. huxleyi* morphotype B/C is considered to be geographically restricted to the Southern Ocean (Cubillos et al., 2007; Cook et al., 2013)

we limit the comparison of our results to studies reporting this morphotype conducted only in the Southern Ocean." (lines 1052-1053).

*R2-C98: Line 676, 682: Salter et al. (2014) missing in the reference list*
**R2-R98**: Corrected according to reviewer 2's suggestion. The reference is now listed in the new version of the manuscript.

*R2-C99: Line 679, 683: Rembauville et al. missing in the reference list*
**R2-R99**: Corrected according to reviewer 2's suggestion. The reference is now listed in the new version of the manuscript.

*R2-C100: Line 698: Analysis of the sediment trap materials>>> analysis of two sediment trap material*

**R2-R100**: This sentence has been rephrased "Analysis of the materials captured by two sediment traps deployed…". Moreover, this sentence has been split into two following reviewer 2's suggestion R2-C6.

---

## Author Response (AR2)

Dear Emilio Marañón,

We greatly appreciate your last comment on the possible impact of light intensity in the seasonality of coccolith weight and length on the observed changes in our traps. We have included this hypothesis in the discussion and corrected the conclusions accordingly. Moreover, two additional changes have been included:

- In the description of morphotype B/C, one of the co-authors suggested to remove the words "often straight or concave (Cubillos et al., 2007)" because this feature is not always observed in this morphotype.
- A short paragraph discounting carbonate chemistry influences has also been included.

Moreover, Please find a new version of the manuscript with tracked changes. All the new changes can be found in lines 397, 601-632, 655, 673-680, 696-713, 819-823 and 844-846. Additionally, the acknowledgement section has been updated.

Many thanks,

Andrés Rigual

[revised manuscript text omitted]